# THE ROLE OF FORGETTING IN FINE-TUNING REINFORCEMENT LEARNING MODELS

## ABSTRACT

Fine-tuning is a widespread technique that allows practitioners to transfer pre-trained capabilities, as recently showcased by the successful applications of foundation models. However, fine-tuning pre-trained reinforcement learning (RL) agents remains a challenge. This work conceptualizes one specific cause of poor transfers in the RL setting: *forgetting of pre-trained capabilities*. Namely, due to the distribution shift between the pre-training and fine-tuning data, the pre-trained model can significantly deteriorate before the agent reaches parts of the state space known by the pre-trained policy. In many cases, re-learning the lost capabilities takes as much time as learning them from scratch. We identify conditions when this problem occurs, perform a thorough analysis, and identify potential solutions. Namely, we propose to counteract deterioration by applying techniques that mitigate forgetting. We experimentally confirm this to be an efficient solution; for example, it allows us to significantly improve the fine-tuning process on Montezuma's Revenge as well as on the challenging NetHack domain.

## 1 INTRODUCTION

Fine-tuning pre-trained neural networks is a widespread technique in deep learning (Yosinski et al., 2014; Girshick et al., 2014). Its power has recently been showcased by spectacular successes in the deployment of foundation models in downstream tasks, including natural language processing (Chung et al., 2022), computer vision (Sandler et al., 2022), automatic speech recognition (Zhang et al., 2022), and cheminformatics (Chithrananda et al., 2020), among others. These successes can mainly be observed in the domains of self-supervised and supervised learning, and the results of a similar caliber have yet to find their way to reinforcement learning (RL).

This work investigates the reasons for the difficulty in fine-tuning RL models. In RL, the agent explores the state space, acquiring data sequentially and learning along the way. This means the policy is ever-changing, and the experience data stream is inherently non-stationary, unlike in self-supervised and supervised learning scenarios. Fine-tuning adds another layer of a distribution shift in the form of acting in an altered environment, performing a new task, or executing a perturbed expert policy. Our analysis reveals that these cause interference in the learning dynamics of neural networks, leading to the deterioration of pre-trained model capabilities. We dub this phenomenon *forgetting of pre-trained capabilities* (FPC).

We study two specific instances of FPC: *state coverage gap* and *imperfect cloning gap*. The first one occurs when the pre-trained agent performs well only on a part of the state space, see Figure 1. The second instance might happen when compounding approximation errors induce significant performance degradation of a pre-trained imitation agent. We start by demonstrating that the phenomena can arise already in a simple two-state MDP. Further, we construct a sequential setup consisting of robotic tasks from Yu et al. (2020); Wołczyk et al. (2021), which let us perform a thorough analysis when using RL algorithms with deep neural networks. Importantly, we show that the forgetting problem can be severe. The pre-trained model often deteriorates completely and we do not any transfer of its capabilities to the new task.

We propose to mitigate the forgetting of pre-trained capabilities problem by applying knowledge retention techniques (Kirkpatrick et al., 2017; Rebuffi et al., 2017; Wołczyk et al., 2021). These approaches preserve previously obtained skills in neural networks while training on new data by

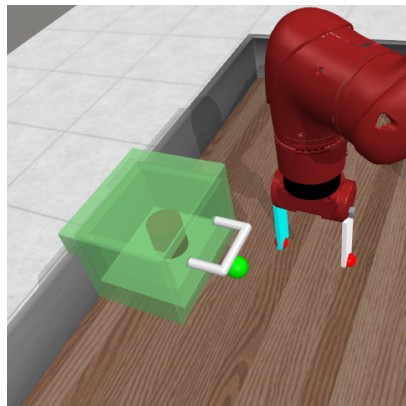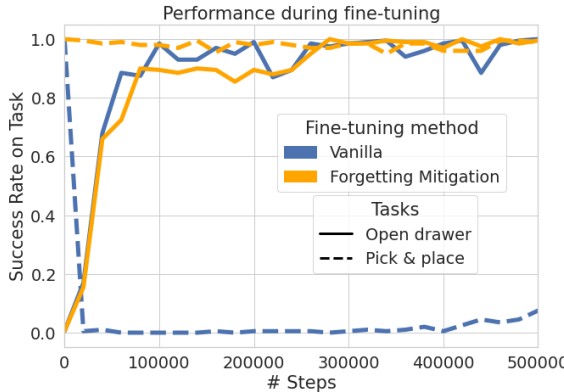

Figure 1: Example of FPC (state coverage gap). We assume that pre-trained model is able to pick and place objects (e.g. the cylinder), however, it does not know how to open drawers. When fine-tuning on a new task, in which the goal is to pick up an object placed inside a closed drawer, we observe an instance of the forgetting of pre-trained capabilities problem. Namely, the model rapidly forgets how to manipulate objects before learning to open the drawer and struggles to reacquire this skill (dashed blue line). Forgetting mitigation techniques alleviate this issue (dashed orange line). At the same time, in both cases, the model learns how to open the drawer (solid lines).

constraining the optimization process. Such methods can significantly improve the fine-tuning performance and outperform models trained from scratch. We demonstrate this by providing results in a sequential robotic environment and in realistic RL domains: NetHack and Montezuma's Revenge.

Our findings go against common practices in supervised transfer learning, which state that the best approach to maximize knowledge transfer to the downstream task is to fine-tune the model without constraints on optimization (Radford et al., 2018; Devlin et al., 2019; Dosovitskiy et al., 2020). Through a set of comprehensive experiments, we show that fine-tuning in RL is different and one should be careful to preserve prior knowledge, even if the only metric of interest is the downstream task performance. Highlighting such potential pitfalls is an important step in the development of strong foundation models in RL.

Our contributions are as follows:

- We pinpoint forgetting of pre-trained capabilities as a critical problem in fine-tuning online RL models and provide a compelling conceptualization of this phenomenon, and its two instances: state coverage gap and imperfect cloning gap.
- We showcase a successful use of knowledge retention techniques which in turn allows us to efficiently leverage the pre-trained model.
- We thoroughly examine the forgetting phenomenon in a simple MDP and robotic sequential environment, and validate finding in realistic RL domains.

## 2 FORGETTING AND INTERFERENCE IN REINFORCEMENT LEARNING

Forgetting of pre-trained capabilities is a consequence of the learning dynamics of neural networks. In particular, due to reusing the same parameters to evaluate different states, learning can lead to interference, where improving the estimation of the value function on one state deteriorates the estimation on a different state, see e.g. Schaul et al. (2019). To illustrate these phenomena, let us split the state space into two sets: CLOSE and FAR. The states in CLOSE are close to the starting state, which the agent frequently sees. The states in FAR are reachable only by going through CLOSE; hence, they are further away from the starting state and can only be reached once some learning on CLOSE happens. In this paper, we consider the following two scenarios.

First, the *state coverage gap*, in which a policy is pre-trained on a different environment and ends up performing well mostly on FAR. The agent using this policy does not know how to behave on CLOSE but must go through it to reach FAR. Consequently, it must learn how to act on CLOSE, and due to the aforementioned interference, its behavior on FAR will deteriorate considerably and will

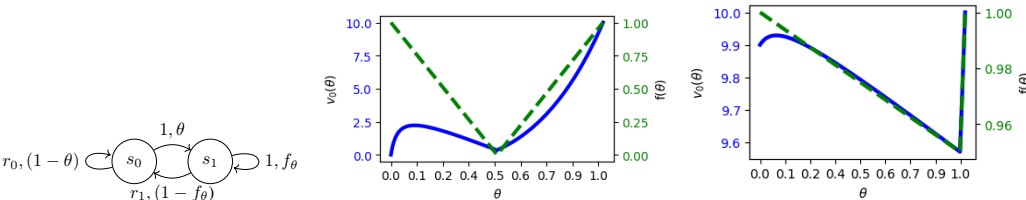

(a) MDP with two states with $\gamma = 0.9$.

(b) Value of $s_0$ (blue) and $f_\theta$ (green) for $r_0 = 0$, $r_1 = -1$, and $f_\theta = 2|\theta - 0.5|$.

(c) Value of $s_0$ (blue) and $f_\theta$ (green) for $r_0 = 0.99$, $r_1 = 0$, $\epsilon = 0.05$, $f_\theta = 1 - \frac{\epsilon\theta}{1-\epsilon/2}$ for $\theta \leq 1 - \epsilon/2$ and $f_\theta = 2\theta - 1$ for $\theta > 1 - \epsilon/2$.

Figure 2: (a) A toy two-state MDP. Each arrow depicts a transition between states, and the annotation encodes the reward and the probability of transition from the policy. (b,c) A policy with its corresponding value function $v_0(\theta)$, for two variants of parameterization and reward functions.

have to be re-acquired. This case is illustrated in Figure 1, with `open drawer` (`pick & place` resp.) being CLOSE (FAR reps.), and is representative of common transfer RL scenarios where the state spaces of upstream and downstream task are similar but not identical (Parisotto et al., 2015; Rusu et al., 2016; 2022)

Second, the *imperfect cloning gap*, in which the pre-trained parameters correspond to a perturbed version of a good policy on the current environment. Such a case frequently arises due to approximation errors when imitating an expert policy or slight changes in the environment between pretraining and fine-tuning. This discrepancy, even if small, can lead to an imbalance where the agent visits states in CLOSE more often than FAR. While trying to correct the slightly suboptimal policy on CLOSE, the policy on FAR can get worse due to interference. We can imagine this issue accumulating over a series of subsets of states progressively more distant from the starting state, leading to a snowball-like effect similar to the initial scenario. This problem represents transfer RL setups, where the model is pre-trained offline on a static dataset and fine-tuned online on the enviornment (Nair et al., 2020; Baker et al., 2022; Zheng et al., 2023).

**Example in 2-state MDP** We now show that the above two scenarios of forgetting of pre-trained capabilities can happen even in a very simple 2-state MDP. This observation fits well into the RL tradition of showing counterexamples on small MDPs (Sutton & Barto, 2018). The MDP, shown in Figure 2(a), consists of two states, labeled as $s_0$ and $s_1$, and an action space which is a singleton. The transition between states is stochastic and is indicated by an arrow annotated by a reward and transition probability. For example, a transition from $s_1$ to $s_0$ happens with probability $1 - f_\theta$ and grants a reward $r_1$. For simplicity, we treat fine-tuning as gradient ascent on the value function until an extreme point is reached. The remaining details can be found in Appendix A.

In Figure 2(b), we present a state coverage gap scenario, where we fine-tune a policy that was pre-trained on a subset of downstream states and we show that it can lead to divergence. Here, we have an MDP where the initial policy $\theta = 0$ was trained only on state $s_1$. Since $f_0 = 1$, such a policy stays in $s_1$ once it starts from $s_1$. If we now try to fine-tune this policy where the starting state is $s_0$, the agent will forget the behavior in $s_1$ due to the interference caused by the parametrization of the policy. This in turn will lead the system to converge to a suboptimal policy $\theta = 0.11$ with a value of 2.22. In this case, the environment has changed by introducing new states that need to be traversed to reach states on which we know how to behave. Learning on these new states that are visited early on will lead to forgetting of the pre-trained behavior.

Subsequently, in Figure 2(c), we provide an example of imperfect cloning gap. In this scenario, $\theta = 1$ (with $f_1 = 1$) represents the optimal behavior of staying in $s_1$ and achieving maximum total discounted returns equal to 10. However, for a given parametrization of $f_\theta$, this maximum can be unstable, and adding a small noise $\epsilon$ to $\theta$ before fine-tuning will lead to divergence towards a local maximum at $\theta = 0.08$ with the corresponding value 9.93. Perturbing $\theta$ by $\epsilon$ will make the system visit $s_0$ more often, and learning on $s_0$ with further push $\theta$ away from 1, forgetting the skill of *moving to and staying in $s_1$*.

**Reinforcement learning (RL)** is formalized using Markov Decision Processes (MDPs) (Sutton & Barto, 2018). An MDP is a tuple $\mathcal{M} = (\mathcal{S}, \mathcal{A}, p, R, \gamma)$, where $\mathcal{S}$ is a state space, $\mathcal{A}$ is an action space, $p : \mathcal{S} \times \mathcal{A} \to P(\mathcal{S})$ is a transition kernel, $R : \mathcal{S} \times \mathcal{A} \to \mathbb{R}$ is a reward function, and $\gamma \in [0, 1]$ is a discount factor. The agent interacts with the environment using a policy $\pi : S \to P(\mathcal{A})$ with the goal of maximizing the expected sum of discounted rewards: $\mathbb{E}_\pi[\sum_{t=0}^{\infty} \gamma^t r_t]$, where $r_t$ is the reward at timestep $t$.

**Knowledge retention** Since we identified forgetting as our main object of study, we consider the following popular methods for knowledge retention from continual learning literature: L2, Elastic Weight Consolidation (EWC), behavioral cloning (BC), and episodic memory (EM). These approaches were tested previously in the continual RL setting (Wołczyk et al., 2021; Kirkpatrick et al., 2017), but were not investigated in the fine-tuning scenario we examine here. The first two are regularization-based approaches, which apply a penalty on parameter changes by introducing an auxiliary loss: $\mathcal{L}_{aux}(\theta) = \sum_i F^i(\theta_{\text{pre}}^i - \theta^i)^2$, where $\theta$ (resp $\theta_{\text{pre}}$) are the weights of the current (resp. pre-trained) model, and $F^i$ are weighting coefficients. For EWC (Kirkpatrick et al., 2017), $F$ is the diagonal of the Fisher matrix, while L2 uses $F^i = 1$ for each $i$. We also use behavioral cloning (BC), an efficient replay-based approach (Rebuffi et al., 2017; Wolczyk et al., 2022). We implement BC in the following way. Before the training, we gather a subset of states $\mathcal{S}_{BC}$ on which the pre-trained model $\pi_{pre}$ was trained, and we construct a buffer $\mathcal{B}_{BC} := \{(s, \pi_{pre}(s)) : s \in \mathcal{S}_{BC}\}$. For the fine-tuning phase, we initialize the policy with $\theta_{pre}$ and we apply an auxiliary loss of the form $\mathcal{L}_{aux}(\theta) = \mathbb{E}_{s \sim \mathcal{B}}[D_{KL}(\pi^*(s) \parallel \pi(s))]$ alongside the RL objective. As for episodic memory for off-policy methods, we simply keep the examples from the pre-trained task in the buffer when training on the new task. For more details about knowledge retention methods, see Appendix C.

## 3    FORGETTING DUE TO STATE COVERAGE GAP

In this section, we empirically test that forgetting impacts the transfer learning in setups with state coverage gap, i.e. previously unseen states appear at the start of a downstream task, and show that forgetting mitigation techniques may alleviate this issue. We use Meta-World Yu et al. (2020) and Montezuma's Revenge as our testbed. The codebase for all experiments is available in the supplementary materials.

### 3.1    A SEQUENCE OF ROBOTIC TASKS

To study the forgetting of pre-trained capabilities phenomenon in a more realistic setting, we develop sequential robotic scenarios based on the Meta-World benchmark (Yu et al., 2020). We evaluate performance and measure forgetting of fine-tuning a pre-trained model (hereafter referred to as **Fine-tuning**) and compare against a baseline of training from a random initialization (hereafter referred to as **From Scratch**).

**RoboticSequence** The agent has to perform the following tasks in sequence *during a single episode*: use a hammer to hammer in a nail (`hammer`), push an object from one specific place to another (`push`), remove a bolt from a wall (`peg-unplug-side`), push an object around a wall (`push-wall`). The next task only begins when the previous one is solved. The first two tasks were chosen to be significantly easier and we assume that the pre-trained agent is perfectly capable of solving the last two (`peg-unplug-side` and `push-wall`). `RoboticSequence` maps to the state coverage gap scenario in Section 2, with (`hammer`, `push`) being CLOSE, and (`peg-unplug-side` and `push-wall`) being FAR. The environments share the state and action spaces but differ in terms of state dynamics and reward functions. For more details (including the training paradigm), see Appendix B.1.

**Performance of fine-tuning** Figure 3 shows the performance throughout training on the whole `RoboticSequence` and on each constituent task. The pre-trained model predictably fails when run on the whole sequence as it cannot solve the two first tasks: (`hammer` and `push`). The fine-tuning does not learn faster than a model trained from scratch, even though at the start of the training it is capable of solving two tasks `peg-unplug-side` and `push-wall`. Inspecting the per-task success rate, we see that although the performance on these two tasks is indeed high at the beginning of training, it drops rapidly after a few initial steps. Moreover, this deterioration is severe, i.e., when the training finally reaches these tasks, the performance grows very slowly, in a similar manner to

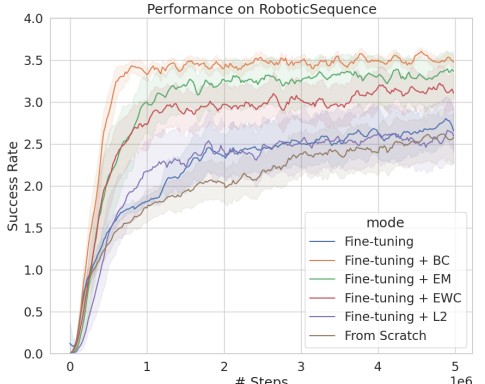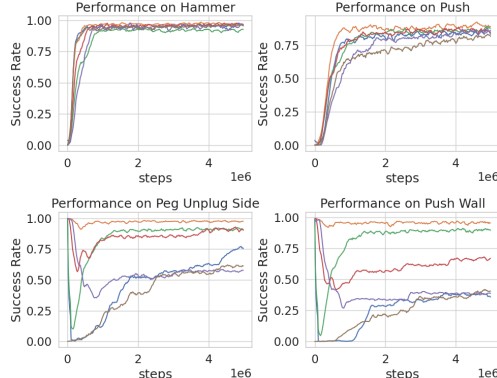

Figure 3: The performance of fine-tuning on a sequential robotic task compared to training from scratch. (Left) The average success of the agent throughout the training, i.e. how many tasks it can solve on average in a single episode. (Right) The success rate of each task evaluated separately. The fine-tuned model rapidly forgets how to solve `peg-unplug-side` and `push-wall`, the tasks it was pre-trained on, and then takes almost as long to relearn them as a model trained from scratch. On the other hand, applying CL methods to limit forgetting unlocks the potential of fine-tuning. BC, EM and EWC are able to maintain or very quickly regain performance on these tasks.

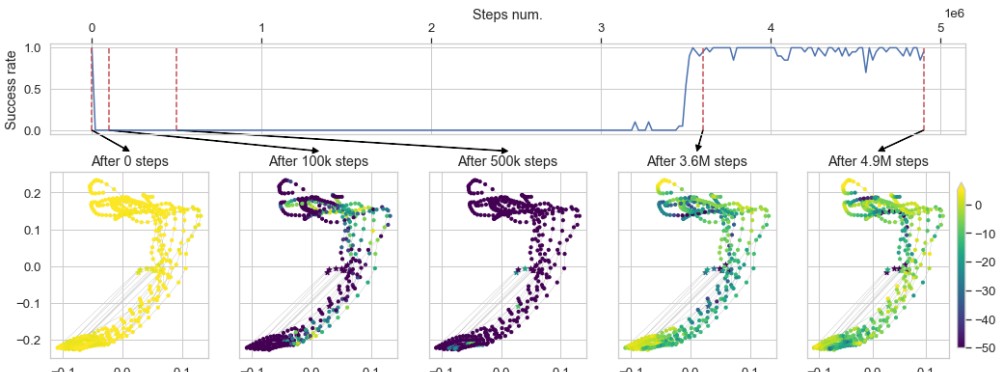

Figure 4: Log-likelihood of the actions of the pre-trained expert policy on `push-wall` as fine-tuning progresses. We gather states from the trajectories of the optimal model, project them to a two-dimensional space using PCA, and color each point using the log-likelihood of the corresponding optimal action under the current policy. As fine-tuning progresses the model forgets the initial solution and is unable to recover it correctly.

a model trained from scratch. As such, the pre-trained model does not exhibit any signs of positive transfer of the knowledge of the last two tasks.

**Application of knowledge retention methods** Results presented in Figure 3 show that the application of the knowledge retention methods can significantly reduce the forgetting of pre-trained capabilities described in the previous paragraph. While behavioral cloning achieves the best results, EWC and episodic memory are still competitive and leverage the pre-trained knowledge better than pure fine-tuning. For these methods, the performance on the pre-trained tasks initially dips, but quickly recovers, which suggests that the relevant knowledge was not lost. The more rudimentary L2 method is able to partly mitigate forgetting, however at the cost of limited plasticity, and it cannot regain the performance on the pre-trained tasks.

**Analysis of forgetting in fine-tuning** In this paragraph we aim to analyze forgetting in a more granular manner. For this purpose, we collect state-action pairs $(s, a^*), a^* \sim \pi_*(s)$ from trajectories sampled with the pre-trained policy $\pi_*$. Throughout the fine-tuning, we measure the log-likelihoods of these actions under the current policy $\pi_\theta$, i.e., $\log \pi_\theta(a^*|s)$ as a measure of forgetting.

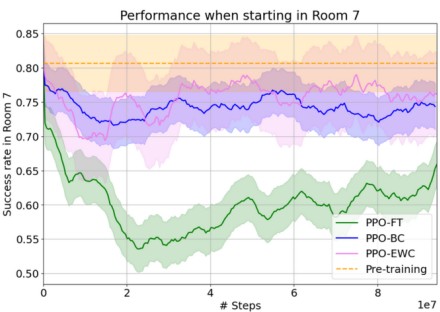
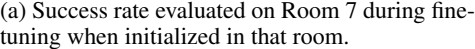
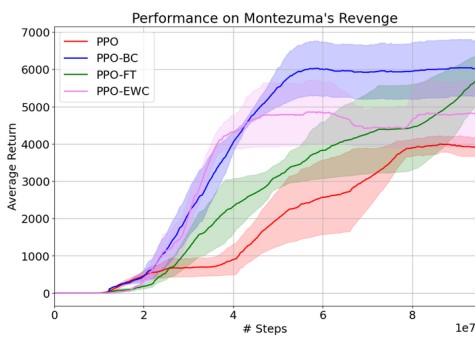

(a) Success rate evaluated on Room 7 during fine-tuning when initialized in that room.

(b) Average return throughout the training. PPO-FT, PPO-EWC and PPO-BC are fine-tuned, while PPO is learned from scratch.

Figure 5: State coverage gap in Montezuma's Revenge. The pre-trained model that only saw rooms from 7th upwards performs significantly better when fine-tuned with knowledge retention.

In Figure 4, we study how the policy deteriorates in certain parts of the state space (projected to 2D using PCA) in the push-wall environment. After $100K$ steps, the model still recalls the correct actions on a part of the state space, but the success rate has already collapsed to $0$, showing that even partial forgetting may significantly impact the performance. After $500K$ steps the likelihood values decline on the whole state space. They increase when the agent relearns the task, but do not reach the original values, showing that the fine-tuned agent learned a different policy.

In Appendix D we provide a more detailed analysis of the forgetting dynamics. In particular, we examine how networks re-learn the forgotten tasks by inspecting changes in the hidden layer representations. We observe that after fine-tuning, the representations in the early layers of the policy network are similar to the pre-trained representations, but the later layers in the policy network and all layers in the critic network retrieve completely different representations. This again suggests that the optimization scheme does not retrieve the pre-trained solution but rather learns a different one from scratch. Additionally, we show that the forgetting problem deepens as we increase the number of unknown tasks, we investigate the role of architecture size, we present results for different sequences of tasks, we attempt to disentangle the impact of policy and representation transfer, and we measure the influence of memory size.

## 3.2 MONTEZUMA'S REVENGE

In this section, we study state coverage gap in Montezuma's Revenge. The main takeaway is that the agent forgets its pre-trained capabilities as the fine-tuning progresses unless a knowledge retention mechanism is applied (PPO-BC), which also happens to improve the agent's overall performance; see Figure 5. More precisely, we assume that during pre-training, we have access to a restricted version of the game that includes only rooms from a certain room onwards (we pick Room 7, denoted on the game map in Figure 10). During fine-tuning, the agent has to solve the whole game, starting from the first room. As such, Room 7 and subsequent ones represent the FAR states, and the preceding rooms represent CLOSE states discussed in Section 2.

We conduct experiments using PPO with Random Network Distillation Burda et al. (2018) to boost exploration (which is essential in this sparse reward environment). We compare three approaches to fine-tuning: vanilla PPO (PPO-FT), PPO with a behavioral cloning loss (PPO-BC) as a knowledge retention method, which uses a dataset of trajectories from the pre-trained policy, and Elastic Weight Consolidation (PPO-EWC). We skip episodic memory, as it requires a replay buffer. Additionally, we test a model trained from scratch as a baseline. In order to monitor forgetting throughout the training, we check their performance in the pre-training scenario, i.e. by spawning them in Room 7. In Figure 5(a) we measure the success rate on this room, defined as the agent successfully leaving the room[1], and we show that indeed as training progresses, performance of PPO-FT falls considerably

---

[1]We use this metrics as the reward signal in Montezuma's revenge is too sparse to provide reliable measurements.

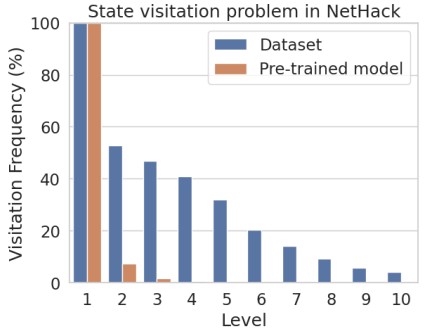

(a) Level visitation in trajectories from the expert and the pre-trained model.

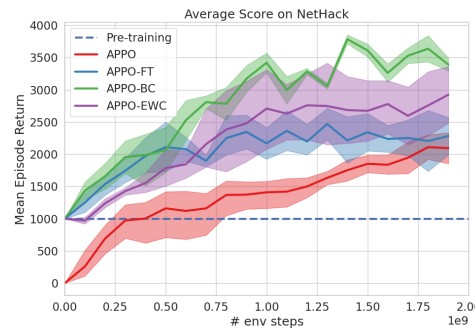

(b) Average return when fine-tuning on NetHack.

Figure 6: Imperfect cloning gap on NetHack. (a) Trajectories gathered online by the pre-trained model differ significantly from those gathered from the rule-based expert since its performance is hard to clone perfectly. (b) As such, the improvements offered by fine-tuning (APPO-FT) over training from scratch (APPO) are diminished, but the performance can be boosted by using a forgetting mitigation technique (APPO-BC).

and only improves after $2M$ steps, which is when the agent revisits Room 7. At the same time, PPO-BC and PPO-EWC maintain good performance in Room 7 throughout the training. Importantly, this difference is reflected in the average return in the game, see Figure 5(b), where PPO-BC manages to learn significantly faster than PPO-FT. PPO-EWC also outperforms training from scratch (PPO) and converges faster than PPO-FT, although it saturates on the lower average return. For more details on Montezuma's Revenge experiments, see Appendix B.3 and for extended analysis see Appendix D.

## 4    FORGETTING DUE TO IMPERFECT CLONING GAP

In this section, we illustrate a common scenario where forgetting of pre-trained capabilities can appear due to the imperfect cloning gap see Section 2. To this end, we use a realistic case of the challenging game of NetHack (Küttler et al., 2020). In particular, we consider the situation when the agent trained on the offline dataset does not perfectly replicate the expert policy used for gathering the dataset. This problem is often present in practice, especially in stochastic environments (like NetHack). This leads to a distribution shift, and consequently, the agent may forget its pre-trained behaviors during fine-tuning. Since this is a common transfer learning scenario, we advocate that fine-tuning methods in RL should carefully consider the impact of forgetting.

NetHack Learning Environment (Küttler et al., 2020) is a rogue-like game consisting of procedurally generated multi-level dungeons. The optimal policy has to encode a variety of behaviors, such as maze navigation, searching for food, fighting, and casting spells, making this a very complex environment without clearly separable subtasks. Since the layout of the dungeon changes in each run, the agent has to learn a general strategy and cannot overfit to a specific map. We use a setup based on Hambro et al. (2022b) who introduce a dataset of labeled trajectories generated from AutoAscend, a rule-based system that achieves state-of-the-art results in NetHack (Hambro et al., 2022a). We use behavioral cloning (BC) policy as our pre-trained model, which is the best-performing offline method tested by Hambro et al. (2022b).

First, we illustrate the distribution shift occurring due to the imperfect cloning gap. In Figure 6(a), we compare the number of NetHack levels visited by the expert and the pre-trained agent. The rule-based expert (AutoAscend) exhibits complex behaviors that rely on a long-term strategy, and as such, the behavioral cloning agent struggles to capture them. In turn, we observe a substantial performance degradation in terms of the number of levels visited. Although the pre-trained agent has significant knowledge of the states beyond the second level, the agent rarely sees this part of the state space at the beginning of fine-tuning. As such, this is a case of imperfect cloning gap, and we hypothesize it can again lead to forgetting of pre-trained capabilities, where the first level corresponds to CLOSE and the subsequent levels correspond to FAR.

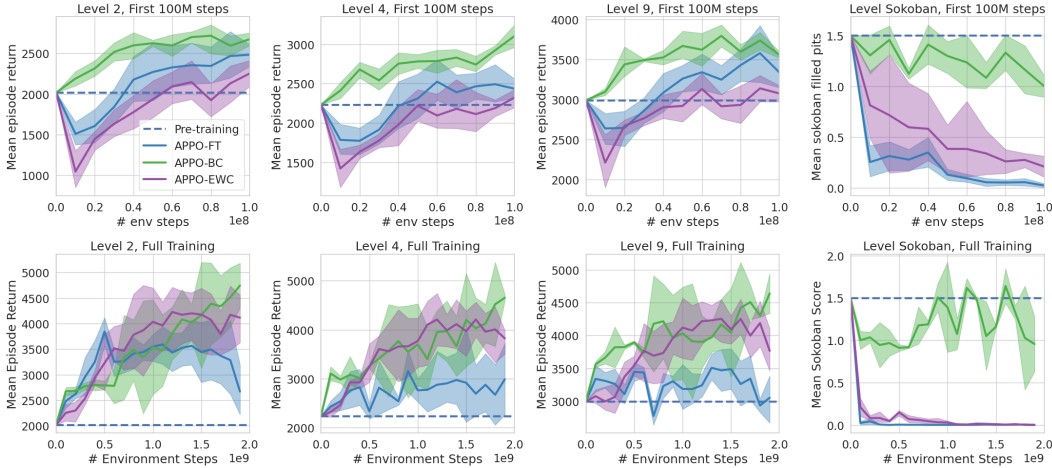

Figure 7: Average return of each of each NetHack method throughout the training, evaluated when started on a specific dungeon level. The top row shows the initial phase of the training (up to 100M steps), and the bottom row shows the full training (up to 2B steps, note the x-axis scale change). For more details about how models are evaluated on specific dungeon levels see Appendix B.2

We test this conjecture by comparing vanilla fine-tuning (APPO-FT) to fine-tuning with an Elastic Weight Consolidation (APPO-EWC) and with a behavioral cloning loss (APPO-BC), which we apply using the states from the AutoAscend pre-training dataset. In all cases, we base on APPO (Petrenko et al., 2020) (following the setup from (Hambro et al., 2022b)) and we also run APPO trained from scratch as a baseline (APPO). Figure 6(b) shows that although at the beginning of training, vanilla fine-tuning outperforms training from scratch, after 2B steps, their performance converges to a similar level. On the other hand, knowledge retention methods (APPO-BC and APPO-EWC) manage to outperform the other baselines, especially in the long run. We hypothesize that only in the later part of the training the agent begins to visit later levels more often, and then keeping good performance on that part of the state space really starts to matter.

To investigate this claim, we look at forgetting in a more fine-grained way. To this end, we gather trajectories from the expert agent, and we extract states $S_\ell$ when it enters level $\ell \in \{2, 4, 9\}$; $|S_\ell| = 200$. Additionally, we gather states from a special NetHack level, which resembles the game of Sokoban $S_s$[2]. We evaluate the performance from the saved states, the rationale being that the agent will not see these states at the start of fine-tuning, and thus we can probe the magnitude of forgetting.

In Figure 7, we look at forgetting at the beginning of the training (top row) as well as over the whole training (bottom row). For the standard dungeon starting states, we observe mild forgetting, as the performance dips at the beginning but quickly recovers. We speculate that this is due to the fact that levels share a lot of similar features and challenges (e.g. killing monsters) and thus there is a positive transfer between earlier levels and subsequent ones. At the same time, we see that the performance of APPO-FT stagnates over time while APPO-BC is still able to increase the score. Additionally, we observe severe forgetting for APPO-FT on $S_s$, without signs of recovery. This is consistent with our hypothesis since Sokoban requires a completely different strategy, and experiences from standard maze exploration will not transfer there. On the other hand, APPO-BC manages to maintain the Sokoban-solving skills by the end of the training. In Appendix F we run additional experiments on applying different types of distillation in this problem, we provide a further analysis of Sokoban results, and we show the plots for all levels along with additional metrics.

## 5 RELATED WORK

**Transfer in RL** Due to high sample complexity and computation costs, training reinforcement learning algorithms from scratch is expensive (Ceron & Castro, 2021; Vinyals et al., 2019; Machado et al.,

---

[2]Sokoban is an NP-hard puzzle where the goal is to push boxes on target locations. This is a popular testing ground for RL algorithms, e.g., Czechowski et al. (2021), see more details in Appendix B.2.

2018a). As such, transfer learning and reusing prior knowledge as much as possible (Agarwal et al., 2022) are becoming more attractive. However, the fine-tuning strategy massively popular in supervised learning (Bommasani et al., 2021; Yosinski et al., 2014; Girshick et al., 2014) is relatively less common in reinforcement learning. Approaches that are often used instead include kickstarting (Schmitt et al., 2018; Lee et al., 2022a), and reusing offline data (Lee et al., 2022b; Kostrikov et al., 2021), skills (Pertsch et al., 2021) or the feature representations (Schwarzer et al., 2021; Stooke et al., 2021). Fine-tuning in RL is often accompanied by knowledge retention mechanisms, even though they are sometimes not described as such. In particular, Baker et al. (2022) includes a regularization term to limit forgetting, Kumar et al. (2022) mixes new data with the old data, and Seo et al. (2022) introduces modularity to the model. Here, we focus on the characterization and the experimental analysis of this issue in fine-tuning RL models, and pinpointing some specific settings when forgetting might occur, such as imperfect cloning gap.

**Continual reinforcement learning** Continual RL deals with learning over a changing stream of tasks represented as MDPs. (Khetarpal et al., 2022; Wołczyk et al., 2021; Nekoei et al., 2021; Powers et al., 2022; Huang et al., 2021; Kessler et al., 2022a). Several works propose methods for continual reinforcement learning based on replay and distillation (Rolnick et al., 2019; Traoré et al., 2019), or modularity (Mendez et al., 2022; Gaya et al., 2022). Although relevant to our study, these works usually investigate changes in the dynamics of non-stationary environments. In this paper, we switch the perspective and focus on the data shifts occurring during fine-tuning in a stationary environment. In fact, some of the standard techniques in RL, such as using the replay buffer, can be seen as a way to tame the non-stationarity inherent to RL (Lin, 1992; Mnih et al., 2013). We highlight an important difference between a standard CL setup and ours. In CL one trains the agent on $N$ environments in sequence: first on task 1, then on task 2, and so on. In our case, we fine-tune the agent on a single environment, where each trajectory can comprise all $N$ tasks. Using Montezuma's Revenge as an example, CL would train the agent for a set number of steps on the first room and then the same number of steps on the next rooms. In our setup, we train the agent to solve the entire game which requires solving (almost) all of the rooms

## 6 CONCLUSIONS

This study investigates the forgetting of pre-trained capabilities in fine-tuning RL models. We show that fine-tuning a pre-trained model on a task where the relevant data is not available at the beginning of the training might lead to a rapid deterioration of the prior knowledge. We highlight two specific cases: state coverage gap and imperfect cloning gap We investigate these phenomena in simple toy settings (two-task MDPs) as well as more realistic simulations (continuous control on a compositional robotic environment), standard RL benchmarks (Montezuma's Revenge) challenging environments that remain unsolved (NetHack).

**Limitations and Societal Impact** Although we aim to comprehensively describe forgetting of pre-trained capabilities, our study is limited in several ways. First of all, we only confirmed that imperfect cloning gap appears with behavioral cloning as the offline pre-training. Although we believe that similar phenomena will appear for other offline RL methods, we did not verify this empirically. Additionally, in our experiments we used fairly simple knowledge retention methods to illustrate the forgetting problem. We believe that CL offers numerous more sophisticated methods that should perform great on this problem (Mallya & Lazebnik, 2018; Ben-Iwhiwhu et al., 2022; Mendez et al., 2022; Khetarpal et al., 2022). We deem this important future work.

Knowledge retention methods can be harmful if the pre-trained policy is suboptimal since they will stop the fine-tuned policy from improving. In our experiments, we do not observe this problem, as we try to only retain knowledge on the parts of the state space where the agent is already proficient, and we try to find the right regularization strength through hyperparameter search. However, in other environments, it might not be easy to identify the part of the state space where the policy should be preserved. We see this as important future work. Finally, knowledge retention methods require additional computation resources and bigger memory in order to apply distillation or regularization. In the standard CL setting complexity of behavioral cloning is linear in memory and quadratic in computation, it remains to be seen how these constraints will transfer to the fine-tuning setting, as the pre-training tasks get more and more complex.

## REPRODUCIBLITY

In order to make our work more reproducible, we carefully describe the experimental setup and the hyperparameters for each domain in Appendix A and B. We carefully discuss the studied methods in Appendix C. Additionally, we include the source code for all experiments in the supplementary materials.

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

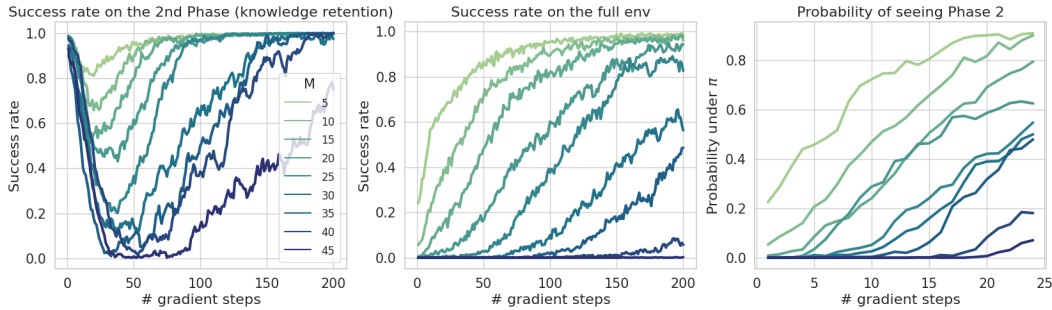

Figure 8: Forgetting of pre-trained capabilities in APPLERETRIEVAL. (Left) Forgetting becomes more problematic as $M$ (the distance from the house to the apple) increases and (center) hinders the overall performance. (Right, note x-scale change) This happens since the probability of reaching Phase 2 in early training decreases.

## A TOY EXAMPLES – MDP AND APPLERETRIEVAL

Consider the 2-state MDP from Figure 2(a). The value of state $s_0$, visualized as a blue line in Figure 2(b) and 2(c), equals

$$v_0(\theta) = \frac{1}{1-\gamma} \frac{\theta + r_0(1-\theta)(1-\gamma f_\theta) + \gamma\theta r_1(1-f_\theta)}{1 - \gamma f_\theta + \gamma\theta}.$$

The policy in Figure 2(c) is defined by $f_\theta = \left(\frac{-\epsilon}{1-\epsilon/2}\theta + 1\right)\mathbf{1}_{\theta \le 1-\epsilon/2} + (2\theta - 1)\mathbf{1}_{\theta > 1-\epsilon/2}$, while the policy in Figure 2(b) is given as $f_\theta = 2|\theta - 0.5|$. In each case, we treat fine-tuning as the process of adjusting $\theta$ towards the gradient direction of $v_0(\theta)$ until a local extremum is encountered.

### A.1 SYNTHETIC EXAMPLE: APPLERETRIEVAL

Additionally, we introduce a synthetic example of an environment exhibiting state coverage gap, dubbed APPLERETRIEVAL. We will show that even a vanilla RL algorithm with linear function approximators shows forgetting of pre-trained capabilities.

APPLERETRIEVAL is a 1D gridworld, consisting of two phases. In Phase 1, starting at home: $x = 0$, the agent has to go to $x = M$ and retrieve an apple, $M \in \mathbb{N}$. In Phase 2, the agent has to go back to $x = 0$. In each phase, the reward is 1 for going in the correct direction and $-1$ otherwise. The observation is $o = [-c]$ in Phase 1 and $o = [c]$ in Phase 2, for some $c \in \mathbb{R}$; i.e. it encodes

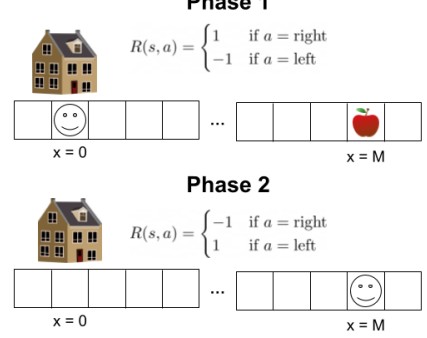

APPLERETRIEVAL environment.

the information about the current phase. Given this observation, it is now trivial to encode the optimal policy: go right in Phase 1 and go left in Phase 2. Episodes are terminated if the solution is reached or after 100 timesteps. Since we can only get to Phase 2 by completing Phase 1, this corresponds to dividing the states to sets $A$ and $B$, as described in Section 2.

We run experiments in APPLERETRIEVAL using the REINFORCE algorithm (Williams, 1992) and assume a simple model in which the probability to move right is given by: $\pi_{w,b}(o) = \sigma(w \cdot o + b), w, b \in \mathbb{R}$. Importantly, we initialize $w, b$ with the weights trained in Phase 2.

**REINFORCE** We use REINFORCE Williams (1992) algorithm for all experiments on APPLERETRIEVAL. We pre-train the single model consisting of two parameters (weight, bias) on the second phase of the environment for 500 episodes and then finetune it on the full environment for 2000 episodes. To reduce noise we update our policy every 10 episodes. During experiments, we used $lr = 0.001$, $\gamma = 0.99$, and $time\_limit = 100$. The algorithm representing REINFORCE construction is presented in Algorithm 1.

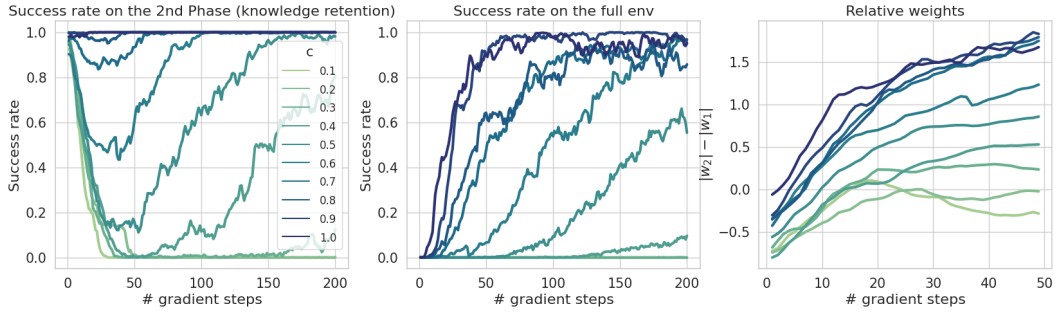

Figure 9: Impact of $c$ on the results for set $M = 30$. For smaller $c$ forgetting (left) is greater and the overall success rate is smaller (center) since it encourages the pre-trained model to find solutions with a high $\frac{|b|}{|w|}$ ratio, as confirmed by looking at weight difference early in fine-tuning (right).

We show experimentally, see Figure 8, that for high enough distance $M$, the forgetting of pre-trained capabilities problem appears. Intuitively, the probability of concluding Phase 1 becomes small enough that the pre-trained Phase 2 policy is forgotten, leading to overall poor performance. In this simple case, we can mechanically analyze this process of forgetting.

Since the linear model in APPLERETRIEVAL has only two parameters (weight $w$, bias $b$) we can analyze and understand what parameter sets lead to forgetting. If the pre-trained policy mostly relies on weight (i.e. $|w| \gg |b|$) then the interference will be limited. However, if the model relies on bias (i.e. $|b| \gg |w|$) then interference will occur as bias will impact the output in the same way in both phases. We can guide the model towards focusing on one or the other by setting the $c$ parameter since the linear model trained with gradient descent will tend towards a solution with a low weight norm. The results presented in Figure 9 confirm our hypothesis, as lower values of $c$ encourage models to rely more on $b$ which leads to forgetting. Such a low-level analysis is infeasible for deep neural networks, but experimental results confirm that interference occurs in practice (Kirkpatrick et al., 2017; Kemker et al., 2018; Ramasesh et al., 2022).

---

**Algorithm 1:** REINFORCE

$i = 0$
**while** $i <$ n_episodes **do**
    state = env.reset()
    log_probs, rewards = [], []
    $j = 0$
    **while** $j <$ update_every **do**
        action, log_prob = model.act(state)
        state, reward, done = env.step(action)
        log_probs.append(log_prob)
        rewards.append(reward)
        **if** done **then**
            state = env.reset()
            $i = i + 1$
            $j = j + 1$
        **end if**
    **end while**
    returns = cumsum(rewards, gamma)
    policy_loss = $-$sum(returns, log_probs)
    optim.zero_grad()
    policy_loss.backward()
    optim.step()
**end while**

---

# B  TECHNICAL DETAILS

## B.1  META WORLD

In this section, we describe the `RoboticSequence` setting used in Section 3.1, and we provide more details about the construction of `RoboticSequence`. The algorithm representing `RoboticSequence` construction is presented in Algorithm 2.

We use multi-layer perceptrons (4 hidden layers, 256 neurons each) as function approximators for the policy and $Q$-value function. For all experiments in this section, we use the Soft Actor-Critic (SAC) algorithm (Haarnoja et al., 2018a). The observation space consists of information about the current robot configuration, see (Yu et al., 2020) for details, and the task ID encoded as a one-hot

vector. In our experiments, we use a pre-trained model that we trained with SAC on the last two tasks (`peg-unplug-side` and `push-wall`) until convergence (i.e. 100% success rate). All experiments on Meta-World are run with at least 20 seeds and we present the results with 90% confidence intervals. We defer further technical details (see Appendix B) and the codebase to the supplementary materials.

In order to make the problem more challenging, we randomly sample the start and goal conditions, similarly as in Wołczyk et al. (2021). Additionally, we change the behavior of the terminal states. In the original paper and codebase, the environments are defined to run indefinitely, but during the training, finite trajectories are sampled (i.e. 200 steps). On the 200th step even though the trajectory ends, SAC receives information that the environment is still going. Effectively, it means that we still bootstrap our Q-value target as if this state was not terminal. This is a common approach for environments with infinite trajectories Pardo et al. (2017).

However, this approach is unintuitive from the perspective of `RoboticSequence`. We would like to go from a given task to the next

---

**Algorithm 2:** STITCHEDENV

**Input:** list of $N$ environments $E_k$, policy $\pi$, time limit $T$.
**Returns:** number of solved environments.
$i = 1; t = 1$ {Initialize env idx, timestep counter}
**while** $i \leq N$ **and** $t \leq T$ **do**
    Take a step in $E_i$ using $\pi$
    **if** $E_i$ is solved **then**
        $i = i + 1; t = 1$ {Move to the next env, reset timestep counter }
    **end if**
**end while**
**return** $i - 1$

---

one at the moment when the success signal appears, without waiting for an arbitrary number of steps. As such, we introduce a change to the environments and terminate the episode in two cases: when the agent succeeds or when the time limit is reached. In both cases, SAC receives a signal that the state was terminal, which means we do not apply bootstrapping in the target Q-value. In order for the MDP to be fully observable, we append the normalized timestep (i.e. the timestep divided by the maximal number of steps in the environment, $T = 200$ in our case) to the state vector. Additionally, when the episode ends with success, we provide the agent with the "remaining" reward it would get until the end of the episode. That is, if the last reward was originally $r_t$, the augmented reward is given by $r'_t = \beta r_t (T - t)$. $\beta = 1.5$ is a coefficient to encourage the agent to succeed. Without the augmented reward there is a risk that the policy would avoid succeeding and terminating the episode, in order to get rewards for a longer period of time.

**SAC** We use the Soft Actor-Critic (Haarnoja et al., 2018a) algorithm for all the experiments on Meta-World and by default use the same architecture as in the Continual World (Wołczyk et al., 2021) paper, which is a 4-layer MLP with 256 neurons each and Leaky-ReLU activations. We apply layer normalization after the first layer. The entropy coefficient is tuned automatically (Haarnoja et al., 2018b). We create a separate output head for each task in the neural networks and then we use the task ID information to choose the correct head. We found that this approach works better than adding the task ID to the observation vector.

For the base SAC, we started with the hyperparameters listed in Wołczyk et al. (2021) and then performed additional hyperparameter tuning. After a quick hyperparameter search, we set the learning rate to $10^{-3}$ and use the Adam Kingma & Ba (2014) optimizer. The batch size is 128 in all experiments. We use L2, EWC, and BC as described in Wołczyk et al. (2021); Wolczyk et al. (2022). For episodic memory, we sample 10k state-action-reward tuples from the pre-trained tasks using the pre-trained policy and we keep them in SAC's replay buffer throughout the training on the downstream task. Since replay buffer is of size 100k, 10% of the buffer is filled with samples from the prior tasks. For each method, we perform a hyperparameter search on method-specific coefficients. Following Wołczyk et al. (2021); Wolczyk et al. (2022) we do not regularize the critic. The final hyperparameters are listed in Table 1.

**CKA** We use Central Kernel Alignment Kornblith et al. (2019) to study similarity of representations. CKA is computed between a pair of matrices, $X \in \mathbb{R}^{n \times p_1}, Y \in \mathbb{R}^{n \times p_2}$, which record, respectively, activations for $p_1$ and $p_2$ neurons for the same $n$ examples. The formula is then given

Table 1: Hyperparameters of knowledge retention methods in Meta-World experiments.

| Method | actor reg. coef. | critic reg. coef. | memory |
|--------|------------------|-------------------|--------|
| L2     | 2                | 0                 | -      |
| EWC    | 100              | 0                 | -      |
| BC     | 1                | 0                 | 10000  |
| EM     | -                | -                 | 10000  |

as follows:

$$\text{CKA}(K, L) = \frac{\text{HSIC}(K, L)}{\sqrt{\text{HSIC}(K, K)\text{HSIC}(L, L)}}, \tag{1}$$

where HSIC is the Hilbert-Schmidt Independence Criterion Gretton et al. (2005), $K_{ij} = k(\mathbf{x}_i, \mathbf{x}_j)$ and $L_{ij} = l(\mathbf{y}_i, \mathbf{y}_j)$, and $k$ and $l$ are two kernels. In our experiments, we simply use a linear kernel in both cases.

**Compute**    For the experiments based on Meta-World, we use CPU acceleration, as the observations and the networks are relatively small and the gains from GPUs are marginal Wołczyk et al. (2021). For each experiment, we use 8 CPU cores and 30GB RAM. The average length of an experiment is 48 hours. During our research for this paper, we ran over 20,000 experiments on Contiual World.

### B.2    NETHACK

**Environment**    NetHack Kenneth Lorber (2023) is a classic and highly complex terminal roguelike game that immerses players in a procedurally generated dungeon crawling experience, navigating through a labyrinth in a world filled with monsters, treasures, and challenges. The NetHack Learning Environment (NLE) introduced in Küttler et al. (2020) is a scalable, procedurally generated, stochastic, rich, and challenging environment aimed to drive long-term research on problems such as exploration, planning, skill acquisition, and language-conditioned RL.

The NLE is characterized by a state space that includes a 2D grid representing the game map and additional information like the player's inventory, health, and other statistics. Thus, the NLE is multimodal and consists of an image, the main map screen, and text. The action space in NLE consists of a set of 120 discrete actions. At the same time, the NLE presents a challenge for RL agents due to its action-chaining behavior. For instance, the player must press three distinct keys in a specific sequence to throw an item, which creates additional complexity to the RL problem. The environmental reward in `score` task, used in this paper, is based on the increase in the in-game score between two-time steps. A complex calculation determines the in-game score. However, during the game's early stages, the score is primarily influenced by factors such as killing monsters and the number of dungeon levels the agent explores. The in-game score is a sensible proxy for incremental progress on NLE. Still, training agents to maximize it is likely not perfectly aligned with solving the game, as expert human players can solve NetHack while keeping the score low.

It is important to note that during training, the agent may not follow levels in a linear sequence due to NetHack's allowance for backtracking or branching to different dungeon parts (as described in `https://nethackwiki.com/wiki/Branch`). Consequently, we can't expect the agent to remember solutions to specific levels, but rather, we aim for it to recall general behavioral patterns for different levels. This highlights the issue of forgetting, even in the absence of strictly defined tasks, contrary to the continual learning literature.

**Dataset**    This paper uses a subset of the NetHack Learning Dataset (NLD) collected by Hambro et al. (2022b) called NLD-AA. It contains over 3 billion state-action-score trajectories and meta-data from 100,000 games collected from the winning bot of the NetHack Challenge Hambro et al. (2022a). In particular, we use about 8000 games of Human Monk. This character was chosen because it was extensively evaluated in the previous work Hambro et al. (2022b) and because the game setup for the Human Monk is relatively straightforward, as it does not require the agent to manage the inventory. The bot is based on the 'AutoAscend' team solution, a symbolic agent that leverages

human knowledge and hand-crafted heuristics to progress in the game. Its decision-making module is based on a behavior tree model.

**Architecture**   We use the solution proposed by the 'Chaotic Dwarven GPT-5' team, which is based on Sample Factory Petrenko et al. (2020) that was also used in Hambro et al. (2022b). This model utilizes an LSTM architecture that incorporates representations from three encoders, which take observations as inputs. The LSTM network's output is then fed into two separate heads: a policy head and a baseline head. The model architecture used both in online and offline settings consists of a joint backbone for both actor and critic. It takes as an input three components: RGB image of the main game screen, `blstats`, and `message`. Where `blstats` refers to the player's status information, such as health and hunger, and `message` refers to the textual information displayed to the player, such as notifications and warnings. These inputs are processed separately using the Nature DQN CNN Mnih et al. (2015) for the image input and fully connected layers for the `blstats` and `message` inputs and are merged before passing to LSTM. This baseline allows for fast training but struggles with learning complex behaviours required for certain roles in the game. Thus, we choose only high-performing Human Monk character. More details about the architecture can be found in Petrenko et al. (2020). The model hyperparameters are shown in Table 2 – analogical to Table 6 from Petrenko et al. (2020).

**Pre-training**   During the offline pre-training phase, we employed Behavioral Cloning (BC) Bain & Sammut (1995); Ross & Bagnell (2010), an imitation learning approach that utilizes a supervised learning objective to train the policy to mimic the actions present in the dataset. To be more specific, it utilizes cross entropy loss function between the policy action distribution and the actions from the NLD-AA dataset. The offline procedure was conducted with the hyperparameters from Table 2, except: `actor_batch_size=256`, `batch_size=128`, `ttyrec_batch_size=512` and `adam_learning_rate=0.001` for two billions steps. For more details on hyperparameters, please refer to the original article Petrenko et al. (2020).

**Fine-tuning**   In the online training phase, we employed a highly parallelizable architecture called Asynchronous Proximal Policy Optimization (APPO) Schulman et al. (2017); Petrenko et al. (2020). In this setup, we can run over 2 billion environment steps under 48 hours of training on A100 Nvidia GPU.

Within the main manuscript, we examined vanilla fine-tuning (APPO-FT) and fine-tuning with a behavioral cloning loss (APPO-BC), the latter is explained in more detail in Appendix C. We used a model pre-trained through offline Behavioral Cloning (BC) training for both fine-tuning approaches. It should be noted that BC does not include a critic, resulting in the baseline head being initialized randomly.

The APPO-BC framework computes the auxiliary loss by utilizing the trajectories generated from the expert (i.e. the AutoAscend algorithm). We scaled the auxiliary losses by a factor of 0.5 (`kickstarting_loss=0.5`).

**Evaluation**   During the evaluation phase, we provide in-game scores and, respectfully, the number of filled pits for Sokoban levels at specific checkpoints during training. Models were evaluated every 100 million environment steps for Figure 6(b) and Figure 7's second row, and every 10 million environment steps for Figure 7's first row. For the per-level evaluation in Figure 7, we employ the AutoAscend expert, used for behavioral cloning in pre-training. We use AutoAscend to play the game and save the state when it reaches the desired level. We generate 200 game saves for each level and evaluate our agents on each save by loading the game, running our agent where the expert finished, and reporting the score our agent achieved on top of the expert's score.

### B.3   MONTEZUMA'S REVENGE

**Environment**   In this section, we provide further details on our experiments with Montezuma's Revenge from Atari Learning Environment (ALE) Machado et al. (2018b). In particular, we investigate forgetting in the scenario in which the agent was pre-trained on deeper rooms of the first level pyramid and then fine-tuned on the whole environment as in the classic gameplay (Figure 10). As

Table 2: Hyperparameters of the model used in NLE. For the most part, we use hyperparameters values from Hambro et al. (2022b).

| Hyperparameter Name | Value |
|---|---|
| activation_function | relu |
| adam_beta1 | 0.9 |
| adam_beta2 | 0.999 |
| adam_eps | 0.0000001 |
| adam_learning_rate | 0.0001 |
| weight_decay | 0.0001 |
| appo_clip_policy | 0.1 |
| appo_clip_baseline | 1.0 |
| baseline_cost | 1 |
| discounting | 0.999 |
| entropy_cost | 0.001 |
| grad_norm_clipping | 4 |
| hidden_dim | 512 |
| batch_size | 128 |
| actor_batch_size | 256 |
| ttyrec_batch_size | 256 |
| penalty_step | 0.0 |
| penalty_time | 0.0 |
| reward_clip | 10 |
| reward_scale | 1 |
| unroll_length | 32 |
| initialisation | 'orthogonal' |
| kickstarting_loss | 0.5 |

well as in the Meta-World setup, after fine-tuning the agent from the regular point of start (room 1), we observe the rapid decline of agent performance in deeper rooms that were mastered during pre-training.

Montezuma's Revenge, released in 1984, presents a challenging platformer scenario where players control the adventurer Panama Joe as he navigates a labyrinthine Aztec temple, solving puzzles and avoiding a variety of deadly obstacles and enemies. What makes Montezuma's Revenge particularly interesting for research purposes is its extreme sparsity of rewards, where meaningful positive feedback is rare and often delayed, posing a significant challenge.

We enumerate rooms according to the progression shown in Figure 10, starting from room 1, where the player begins gameplay. As a successful completion of the room in Figure 5(a), we consider achieving at least one of the following options: either earn a coin as a reward, acquire a new item, or exit the room through a different passage than the one we entered through.

**Architecture** In the experiments, we use a PPO agent with a Random Network Distillation (RND) mechanism Burda et al. (2018) for exploration boost. It achieves this by employing two neural networks: a randomly initialized target network and a prediction network. Both networks receive observation as an input and return a vector with size 512. The prediction network is trained to predict the random outputs generated by the target network. During interaction with the environment, the prediction network assesses the novelty of states, prioritizing exploration in less predictable regions. States for which the prediction network's predictions deviate significantly from the random targets are considered novel and are prioritized for exploration. Detailed hyperparameter values can be found in Table 3.

**Dataset** For behavioural cloning purposes, we collected more than 500 trajectories sampled from a pre-trained PPO agent with RND that achieved an episode cumulative reward of around 7000. In Figure 11 we show the impact of different values of the Kullback–Leibler weight coefficient on agent performance.

Table 3: Hyperparameters of the model used in Montezuma's Revenge. For the most part, we use hyperparameter values from Burda et al. (2018). We used PyTorch implementation by *jcwleo* from https://github.com/jcwleo/random-network-distillation-pytorch

| Hyperparameter Name | Value |
|---|---|
| MaxStepPerEpisode | 4500 |
| ExtCoef | 2.0 |
| LearningRate | 1e-4 |
| NumEnv | 128 |
| NumStep | 128 |
| Gamma | 0.999 |
| IntGamma | 0.99 |
| Lambda | 0.95 |
| StableEps | 1e-8 |
| StateStackSize | 4 |
| PreProcHeight | 84 |
| ProProcWidth | 84 |
| UseGAE | True |
| UseGPU | True |
| UseNorm | False |
| UseNoisyNet | False |
| ClipGradNorm | 0.5 |
| Entropy | 0.001 |
| Epoch | 4 |
| MiniBatch | 4 |
| PPOEps | 0.1 |
| IntCoef | 1.0 |
| StickyAction | True |
| ActionProb | 0.25 |
| UpdateProportion | 0.25 |
| LifeDone | False |
| ObsNormStep | 50 |

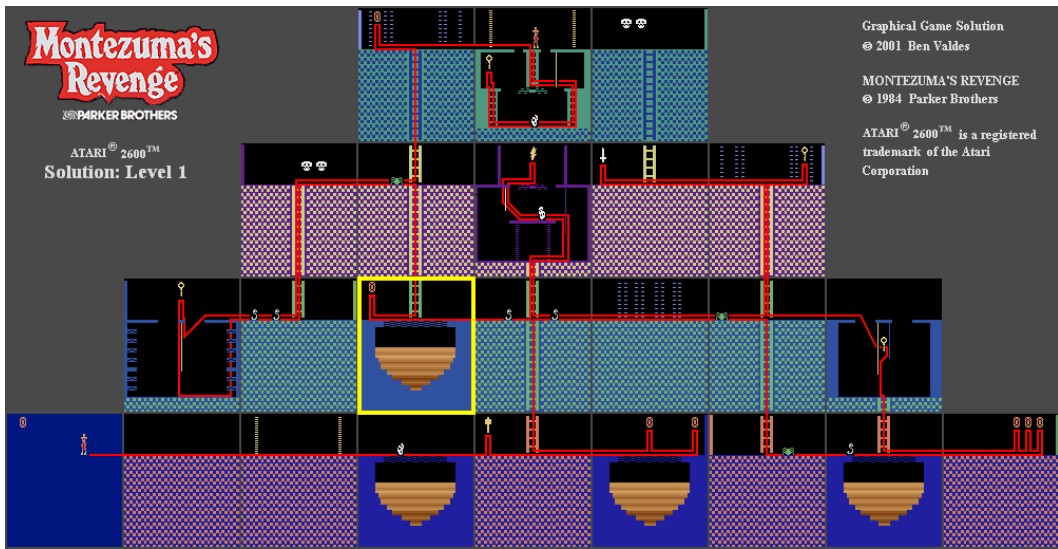

Figure 10: The order in which rooms are visited to complete the first level of Montezuma's Revenge is presented with the red line. We highlight Room 7 using a yellow border. Source: https://pitfallharry.tripod.com/MapRoom/MontezumasRevengeLvl1.html

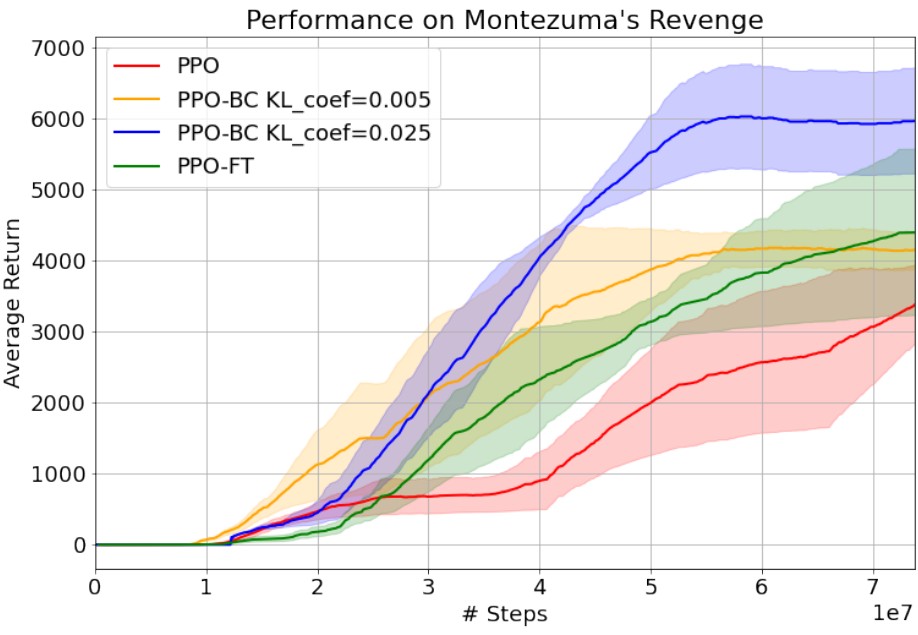

Figure 11: Average return in Montezuma's Revenge for PPO, fine-tuned PPO and two different coefficients for behavioural cloning PPO.

## C  KNOWLEDGE RETENTION METHODS

In this section, we provide more details about the knowledge retention methods used in the experiments, and we briefly describe different types of possible approaches.

In this paper, we mostly focus on forgetting the pre-trained knowledge when fine-tuning only on a single stationary task. However, in continual learning literature that often focuses on the problem of mitigating forgetting, the goal is to usually deal with a sequence of tasks (up to several hundred Lesort et al. (2022)) and efficiently accumulate knowledge over the whole sequence. As such, although here we will describe CL methods with two tasks (corresponding to pre-training and fine-tuning), in practice dealing with a longer sequence of tasks might require more careful considerations.

## C.1 REGULARIZATION-BASED METHODS

Regularization-based methods in CL aim to limit forgetting by penalizing changes in parameters that are relevant to the current task. In particular, a few regularization methods Kirkpatrick et al. (2017); Aljundi et al. (2018) add an auxiliary loss of the following form:

$$\mathcal{L}_{aux}(\theta) = \sum_i F^i(\theta^i_{\text{pre}} - \theta^i)^2,$$ (2)

where $\theta$ are the weights of the current model, $\theta_{\text{pre}}$ are the weights of a prior model, and $F^i$ are weighting coefficients. For **L2** (Kirkpatrick et al., 2017), where we simply want to minimize the L2 norm between the current and the previous solution. In **Elastic Weight Consolidation (EWC)** (Kirkpatrick et al., 2017), $F$ is the diagonal of the Fisher Information Matrix, see (Wołczyk et al., 2021) for details about its implementation in Soft Actor-Critic.

## C.2 DISTILLATION-BASED METHODS

In this work, we use the behavioral cloning approach used previously in continual reinforcement learning setup Wolczyk et al. (2022); Rolnick et al. (2019) This approach is based on minimizing the Kullback-Leibler of action distributions under particular states $D^s_{KL}(p \parallel q) = \mathbb{E}_{a \sim p(\cdot|s)}\left[\log(\frac{p(a|s)}{q(a|s)})\right]$. Assume that $\pi_\theta$ is the current policy parameterized by $\theta$ (student) and $\pi_*$ is the pre-trained policy (teacher).

In behavioral cloning, we apply the following loss:

$$\mathcal{L}_{BC}(\theta) = \mathbb{E}_{s \sim \mathcal{B}}[D^s_{KL}(\pi_\theta \parallel \pi_*)],$$ (3)

where $\mathcal{B}$ is a buffer of data containing states from pre-training. Since we would like to remember how to behave on the entirety of the state space, including the states that are possibly hard to reach (i.e. FAR as described in Section 2), we should make sure these states are included in the loss. This amounts to using states that were used in pre-training, either the offline dataset in the imperfect cloning gap scenario or from the pre-training task in the state coverage gap scenario.

Additionally, the paper introducing the offline dataset for NetHack Learning Environment Hambro et al. (2022b) also introduces a method called "Behavioral Cloning from Observations" (BCO). The version tested in this work differs slightly from their implementation and we describe the differences here. First of all, since we only use the labeled AutoAscend dataset, there is no need to train and use an inverse dynamics model. Second, BCO introduced in Hambro et al. (2022b) uses the auxiliary loss of the following form:

$$\mathcal{L}_{BCO}(\theta) = \mathbb{E}_{s \sim \mathcal{D}_{AA}}[D^s_{KL}(\pi_\theta \parallel \pi_{AA})],$$ (4)

where $\mathcal{D}_{AA}$ denotes the NLD-AA dataset gathered by the AutoAscend bot and $\pi_{AA}$ is the AutoAscend's policy. As such, the teacher here is not the pre-trained model but in fact, the AutoAscend bot. Although the pre-training scheme aims to mimic the AutoAscend bot as closely as possible, the two will in practice, differ, which means that behavioral cloning in this paper might work slightly differently than behavioral cloning from Hambro et al. (2022b). For example, the AutoAscend dataset only contains single actions (i.e. in the action distribution, the whole probability mass lies on a single action), while the pre-trained agent will have a smoother action distribution.

## C.3 NOTE ON CRITIC REGULARIZATION

In actor-critic architectures popular in reinforcement learning, one can decide whether to apply knowledge retention methods only to the actor and only to the critic. If all we care about is the

policy being able to correctly execute the policies for the previous tasks, then it is enough to force the actor to not forget. Since the critic is only used for training, forgetting in the critic will not directly impact the performance. On the other hand, in principle preserving knowledge in the critic might allow us to efficiently re-train on any of the prior tasks. In this paper, following Wolczyk et al. (2022) we focus on regularizing only the actor, i.e. we do not apply any distillation loss on the critic in distillation-based methods and we do not minimize the distance on the L2 norm on the critic-specific parameters.

## C.4 REPLAY-BASED METHODS

A simple way to mitigate forgetting is to add the prior data to the training dataset for the current dataset (in supervised learning (Chaudhry et al., 2019; Buzzega et al., 2021)) or to the replay buffer (in off-policy RL (Rolnick et al., 2019; Kessler et al., 2022b)). By mixing the data from the previous and the current task, one approximates the perfectly mixed i.i.d. data distribution, thus going closer to continual learning.

In our experiments, we use a simple episodic memory (EM) approach along iwth the off-policy SAC algortihm. At the start of the training, we gather a set of trajectories from the pre-trained enviornment and we use them to populate SAC's replay buffer. In our experiments, old samples take 10% of the whole buffer size. Then, throughout the training we protect that part of the buffer, i.e. we do not allow the data from the pre-trained task to be overriden.

Although episodic memory performs well in our experiments, it is difficult to use this strategy in settings with on-policy algorithms. In particular, we cannot trivially use it with PPO in Montezuma's Revenge and with APPO in NetHack as these methods do not use a replay buffer and might become unstable when trained with off-policy data. Additionally, we note that episodic memory seems to work poorly with SAC in traditional continual learning settings Wołczyk et al. (2021); Wolczyk et al. (2022). As such, we focus on the distillation approaches instead.

## C.5 PARAMETER-ISOLATION METHODS

Standard taxonomies of continual learning De Lange et al. (2021) also consider parameter isolation-based (or modularity-based) method. Such methods assign a subset of parameters to each task and preserve the performance by keeping these weights frozen. For example, Progressive Networks Rusu et al. (2016) introduces a new set of parameters with each introduced task, and PackNet Mallya & Lazebnik (2018) freezes a subset of existing weights after each task. Recent works showed that by carefully combining the modules, one can achieve a significant knowledge transfer without any forgetting Veniat et al. (2021); Ostapenko et al. (2021). However, in most cases, methods in this family require access to the task ID. Although we provide the task ID in our controlled `RoboticSequence` environments, most realistic problems, such as NetHack, do not have clearly separable tasks and as such application of such methods to the general fine-tuning problem might be non-trivial.

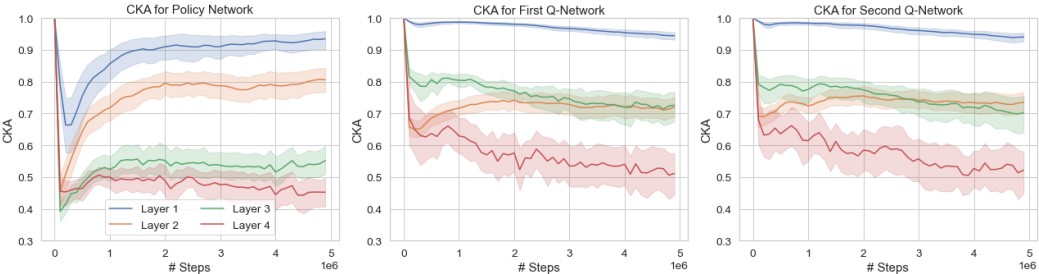

Figure 12: The CKA values throughout vanilla fine-tuning (without knowledge retention methods), computed between the activations of the pre-trained model and the activations of the current model. The higher the values, the more similar the representations.

## D  ANALYSIS OF FORGETTING IN ROBOTIC MANIPULATION TASKS

In this section, we present additional results for our robotic manipulation experiments based on Meta-World.

We use the experimental setting from Section 3.1 and the same sequence of tasks. We adopt the forward transfer metric used previously in Wołczyk et al. (2021); Bornschein et al. (2022) to measure how much pre-training helps during fine-tuning, which serves us also as a proxy of knowledge retention:

$$\text{Forward Transfer} := \frac{\text{AUC} - \text{AUC}^b}{1 - \text{AUC}^b}, \quad \text{AUC} := \frac{1}{T}\int_0^T p(t)\mathrm{d}t, \quad \text{AUC}^b := \frac{1}{T}\int_0^T p^b(t)\mathrm{d}t,$$

where $p(t)$ is the success rate of the pre-trained model at time $t$, $p^b$ denotes the success rate of a network trained from scratch, and $T$ is the training length. By default, we will compute forward transfer on the whole `RoboticSequence`, but it can be also computed on any of its constituent tasks..

**Analysis of internal representations** We examine how activations of the actor and critic networks in SAC change throughout fine-tuning when we do not use any CL methods, with the goal of pinpointing the structure of forgetting. In order to measure the representation shift in the network, we use the Central Kernel Alignment (CKA) (Kornblith et al., 2019) metric, which was previously used in studying forgetting in the supervised learning paradigm (Ramasesh et al., 2020; Mirzadeh et al., 2022), see Appendix B for more details. Before starting the fine-tuning process, we collect optimal trajectories from the pre-trained model along with the activations of the networks after each layer. Then, at multiple points throughout the training process, we feed the same trajectories through the fine-tuned network and compare its activations to the prior activations using CKA. Figure 12 shows that, in general, later layers change more than the early layers, which is consistent with previous studies (Ramasesh et al., 2020). This is particularly visible in the policy network, while the tendency is not as strong for the critic networks, suggesting that the TD-learning guiding the critic leads to different representation learning dynamics.

In the policy network, representations in the early layers change rapidly at the beginning of the fine-tuning process. Then, interestingly, as we solve the new tasks and revisit the tasks from pretraining, CKA increases and the activations become more similar to the pre-trained ones. As such, the re-learning visible in per-task success rates in Figure 3 is also reflected in the CKA here. This phenomenon does not hold for the later layers in the policy network or the $Q$-networks. This suggests that although we are able to retrieve similar representations, the solution we find is significantly different.

**Impact of the network size** Previous studies in supervised continual learning showed that forgetting might become smaller as we increase the size of the neural network Ramasesh et al. (2022); Mirzadeh et al. (2022), and here we investigate the same point in RL using our `RoboticSequence` setting. We run a grid of experiments with hidden dimensions in $\{256, 512, 1024\}$ and number of layers in $\{2, 3, 4\}$. For each of these combinations, we repeat the experiment from Section 3.1 namely, we measure how fine-tuning from a pre-trained solution

compares to starting from random initialization and how the results change when we apply continual learning methods. We omit L2 here, as we find EWC, which is conceptually similar, outperforms it in all cases. The summary of the results is presented in Table 4.

The results do not show any clear correlations between the network size and forgetting, hinting at more complex interactions than these previously showed in continual supervised learning literature Ramasesh et al. (2022). The fine-tuning approach fails to achieve a significant positive transfer for two or four layers, but it does show signs of knowledge retention with three layers. Inspection of the detailed results for the three-layer case, see Figure 18, shows that the fine-tuning performance on the known tasks still falls to zero at the beginning, but it can regain performance relatively quickly. As for the CL methods, we observe that behavioral cloning performs well independently of the size of the network. On the other hand, EWC tends to fail with two layers. Since EWC directly penalizes changes in the parameters, we hypothesize that with a small, two-layer network, the resulting loss of plasticity makes it especially difficult to learn.

**Impact of the number of unknown tasks**   In our APPLERETRIEVAL experiments, we showed that forgetting of pre-trained capabilities is more visible as we increase the amount of time spent before visiting the known part of the state space. We investigate the same question in the context of robotic manipulation tasks by changing the number of new tasks the agent has to solve prior to reaching the ones it was pre-trained on. That is, we study `RoboticSequences` where the last two tasks are `peg-unplug-side` and `push-wall`, as previously, but the first tasks are taken as different length suffixes of `window-close`, `faucet-close`, `hammer`, `push` We call the tasks preceding the pre-trained tasks the *prefix tasks*.

We investigate how the number of the prefix tasks impacts the performance on the known tasks during the fine-tuning process. Table 5 shows the forward transfer metric computed on the pre-trained tasks for fine-tuning, EWC and BC. As the number of prefix tasks grows, the forward transfer values for fine-tuning become smaller, which means that the gains offered by the prior knowledge vanish. Interestingly, even with a single prefix task the forward transfer is relatively low. On the other hand, continual learning methods do not suffer as much from this issue. BC achieves high forward transfer regardless of the setting and EWC experiences only small deterioration as we increase the number of prefix tasks.

**Impact of representation vs policy on transfer**   Although we see significant positive transfer once the forgetting problem is addressed, it remains an open question where this impact comes from. Although there are several studies on the impact of representation learning on transfer in supervised learning (Neyshabur et al., 2020; Kornblith et al., 2021), the same question in RL remains relatively understudied. Here, we try to understand the impact of representation and policy on transfer by resetting the last layer of the network before starting the training. As such, the policy at the beginning is random even on the tasks known from pre-training, but has features relevant to solving these tasks. The improvements should then only come from the transfer of representation.

The results for these experiments are presented in Figure 13. First of all, we observe that, expectedly, this setting is significantly harder, as all methods perform worse than without resetting the head. However, we still observe significant transfer for BC and EWC as they train faster than a randomly initialized model. At the same time, fine-tuning in the end manages to match the performance of BC and EWC, however at a much slower pace. We hypothesize that the gap between knowledge retention methods and fine-tuning is smaller, since now the methods have to re-learn a new policy rather than maintain the old one. This preliminary experiment suggests that the benefits of fine-tuning come from both the policy and the representation since we can still observe a significant, although reduced, transfer after resetting the heads. Maximizing transfer from the representation remains an interesting open question.

**Other sequences**   In order to provide another testbed for our investigations, we repeat the main experiments on another sequence of tasks, namely `shelf-place`, `push-back`, `window-close`, `door-close`, where again we fine-tune a model that was pre-trained on the last two tasks. The results are presented in Figure 16. We find that the main conclusions from the other sequence hold here, although, interestingly, the performance of EWC is significantly better. Additionally, we run experiments on a simple, two task `RoboticSequence` with `drawer-open` and `pick-place`,

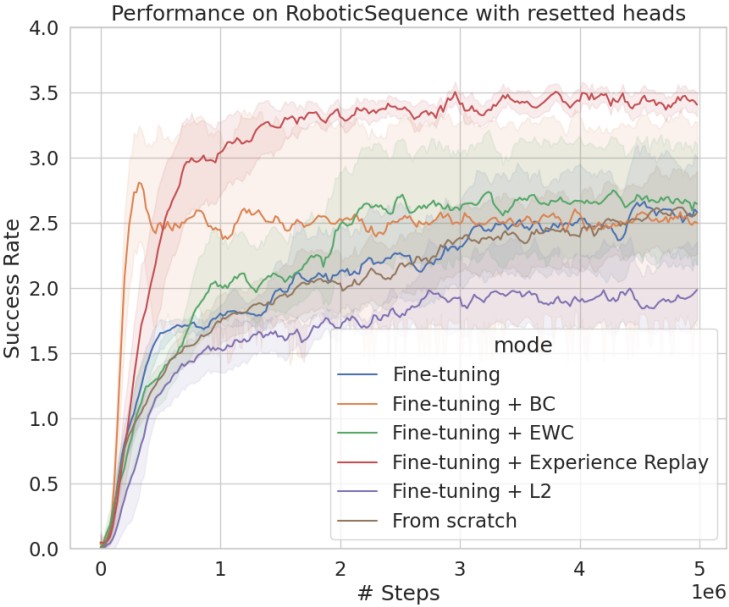

Figure 13: Performance of different methods on the RoboticSequence where we reset the last layer of the policy and critic networks. The results are worse than in the standard case, but there is still some positive transfer, suggesting that benefits come from reusing both the representations as well as the policy.

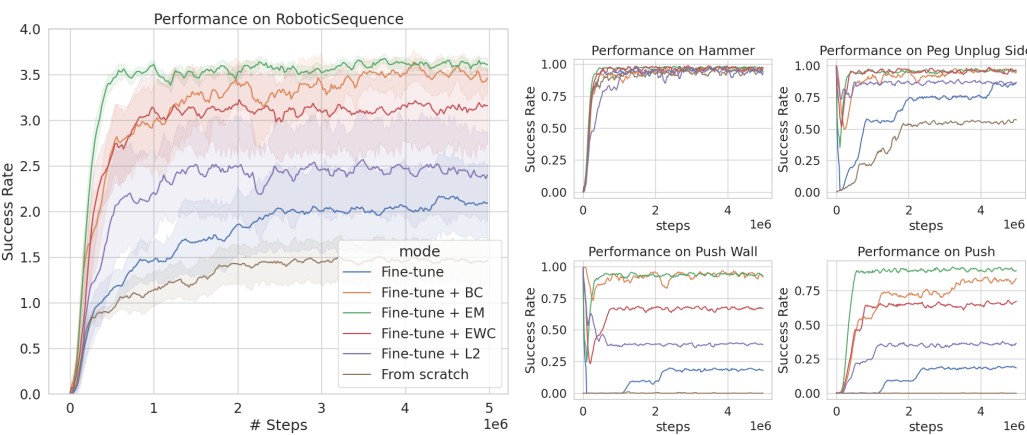

Figure 14: The performance on a robotic sequence where the known tasks are in the middle.

showcased in Figure 1. We used behavioral cloning as an example of a method that mitigates forgetting.

Additionally, we check what happens when the known tasks are "in the middle" of two known tasks. That is, we use the environment consisting of the following sequence of goals: `hammer, peg-unplug-side, push-wall, push` with a model pre-trained on `peg-unplug-side, push-wall`. With this setup, we are especially interested in the impact of different methods on the performance on the last task, i.e. can we still learn new things after visiting a known part of the state space?

The results presented in Figure 14 show that the relative performance of all methods is the same as in our original ordering, however, we observe that EWC almost matches the score of BC. The learning benefits on the last task, `push`, is somewhat difficult to estimate. That is since BC manages to maintain good performance on tasks `peg-unplug-side` and `push-wall`, it sees data from

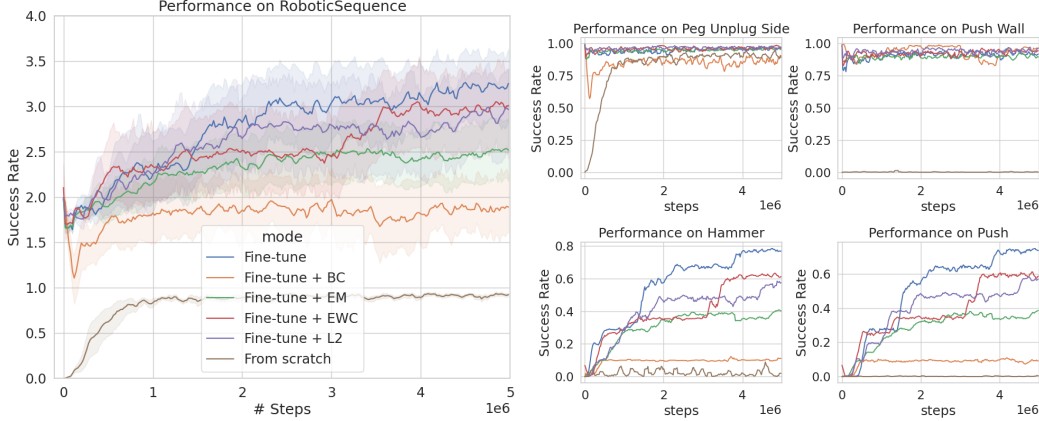

Figure 15: The performance on a robotic sequence where the known tasks are positioned at the beginning.

`push` much sooner than approaches that have to re-learn tasks 2 and 3. However, we observe that even after encountering the later tasks, knowledge retention methods perform much better on `push` than vanilla fine-tuning, which in turn is better than a model trained from scratch.

Finally, we verify that the gap between vanilla fine-tuning and knowledge retention methods does not appear when the relevant skills are only needed at the start of the downstream task. To do this, we use the following sequence of goals: `peg-unplug-side`, `push-wall`, `hammer`, `push` with a model pre-trained on `peg-unplug-side`, `push-wall`. Results in Figure 15 show that indeed in this scenario there is no forgetting and fine-tuning manages just as well or sometimes even slightly better than knowledge retention methods.

**Impact of the memory size on the results**    The memory overhead is an important consideration in fine-tuning with a behavioral cloning loss. We run experiments to check how many samples we actually need to protect knowledge of the previous tasks. Results presented in Figure 17 show that even with 100 samples we are able to keep good performance, at the cost of a higher performance drop on the pre-trained tasks at the beginning of the fine-tuning process.

**Success rate plots for different architectures**    In Figure 18, we present the full results from Table 4 with success rate plots throughout the training.

Table 5: Forward transfer on the pre-trained tasks depending on the number of prefix tasks in `RoboticSequence`.

| Prefix Len | push-wall | | | peg-unplug-side | | |
|---|---|---|---|---|---|---|
| | FT | EWC | BC | FT | EWC | BC |
| 1 | 0.18 [-0.19, 0.43] | 0.88 [0.84, 0.91] | 0.93 [0.89, 0.96] | 0.28 [0.01, 0.46] | 0.77 [0.58, 0.88] | 0.92 [0.88, 0.94] |
| 2 | 0.17 [-0.21, 0.44] | 0.65 [0.44, 0.82] | 0.97 [0.97, 0.98] | 0.15 [-0.08, 0.35] | 0.55 [0.37, 0.70] | 0.95 [0.94, 0.96] |
| 3 | 0.10 [-0.03, 0.23] | 0.64 [0.50, 0.75] | 0.98 [0.98, 0.98] | 0.03 [0.00, 0.06] | 0.41 [0.28, 0.54] | 0.95 [0.95, 0.95] |
| 4 | -0.00 [-0.16, 0.10] | 0.62 [0.48, 0.75] | 0.97 [0.97, 0.98] | 0.03 [-0.00, 0.08] | 0.46 [0.33, 0.59] | 0.94 [0.94, 0.95] |

Table 4: Forward transfer depending on the network architecture, with 90% bootstrap confidence intervals.

| Num. Layers | | 2 | |
| Hidden Dim. | FT | EWC | BC |
| --- | --- | --- | --- |
| 256 | 0.01 [-0.04, 0.06] | 0.01 [-0.16, 0.20] | 0.65 [0.61, 0.69] |
| 512 | 0.10 [0.06, 0.14] | 0.32 [-0.07, 0.68] | 0.73 [0.71, 0.75] |
| 1024 | 0.05 [-0.07, 0.16] | -0.48 [-0.74, -0.23] | 0.59 [0.43, 0.70] |

| Num. Layers | | 3 | |
| Hidden Dim. | FT | EWC | BC |
| --- | --- | --- | --- |
| 256 | 0.20 [0.12, 0.28] | 0.56 [0.38, 0.71] | 0.73 [0.71, 0.75] |
| 512 | 0.35 [0.27, 0.42] | 0.50 [0.44, 0.56] | 0.57 [0.52, 0.60] |
| 1024 | 0.31 [0.24, 0.37] | 0.40 [0.30, 0.49] | 0.53 [0.49, 0.57] |

| Num. Layers | | 4 | |
| Hidden Dim. | FT | EWC | BC |
| --- | --- | --- | --- |
| 256 | 0.07 [-0.06, 0.20] | 0.34 [0.15, 0.51] | 0.62 [0.57, 0.66] |
| 512 | -0.18 [-0.38, -0.02] | 0.43 [0.24, 0.57] | 0.55 [0.49, 0.60] |
| 1024 | -0.10 [-0.33, 0.08] | 0.30 [-0.06, 0.61] | 0.69 [0.63, 0.75] |

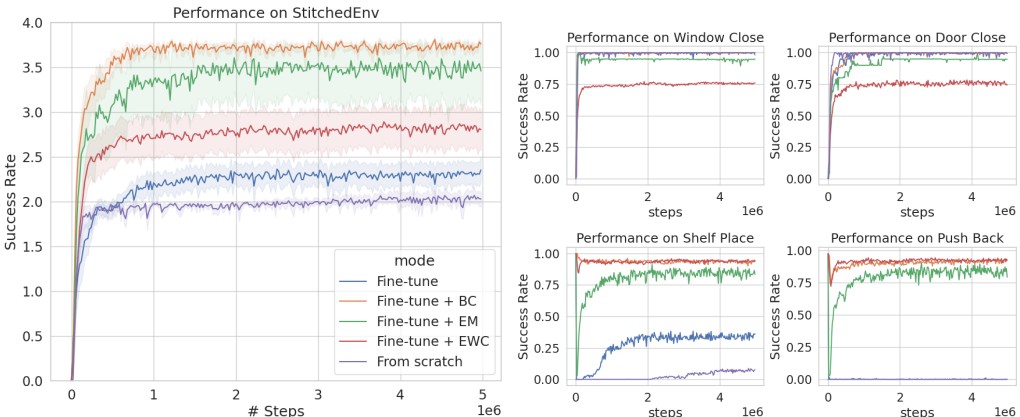

Figure 16: The performance of the fine-tuned model on `RoboticSequence` compared to a model trained from scratch and knowledge retention methods on the sequence `shelf-place`, `push-back`, `window-close`, `door-close`.

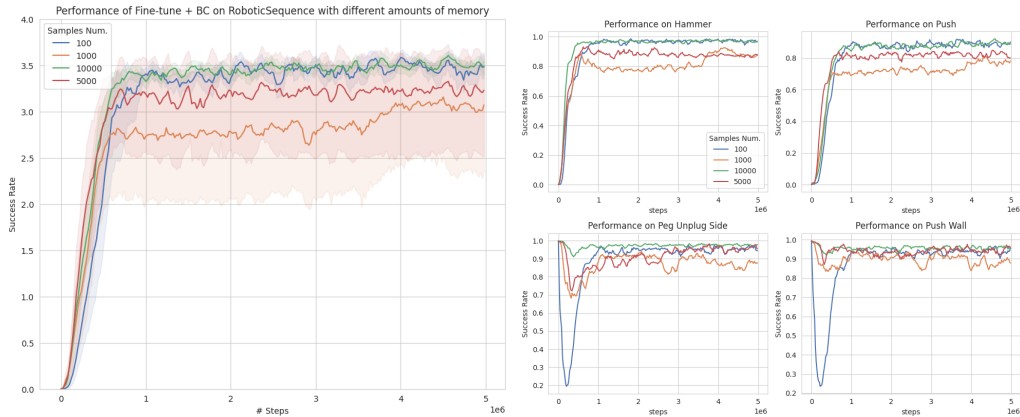

Figure 17: The performance of Fine-tune + BC with different memory sizes. Even with 100 samples we are able to retain the knowledge required to make progress in the training.

Figure 18: Training performance for different architecture choices.

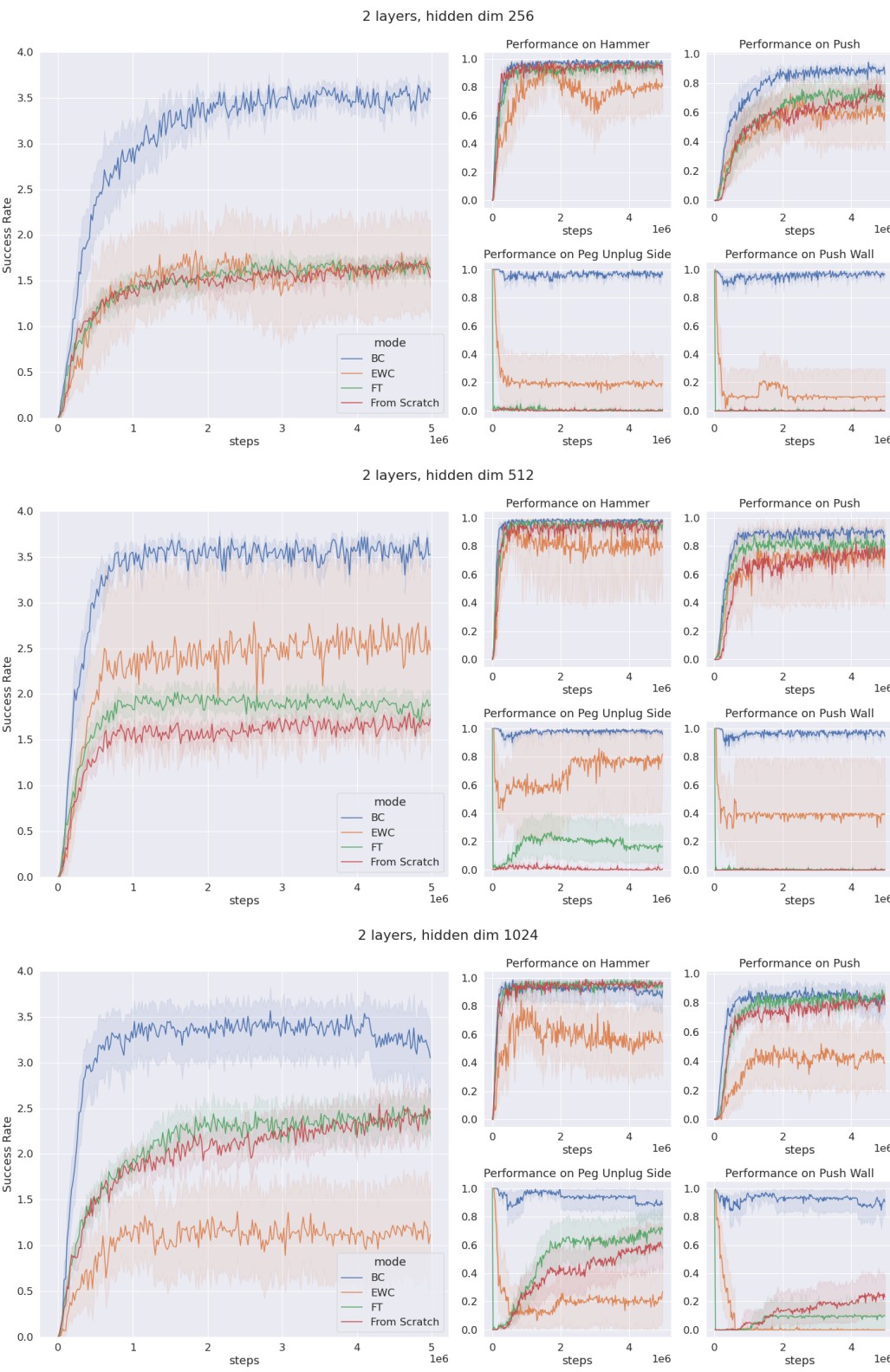

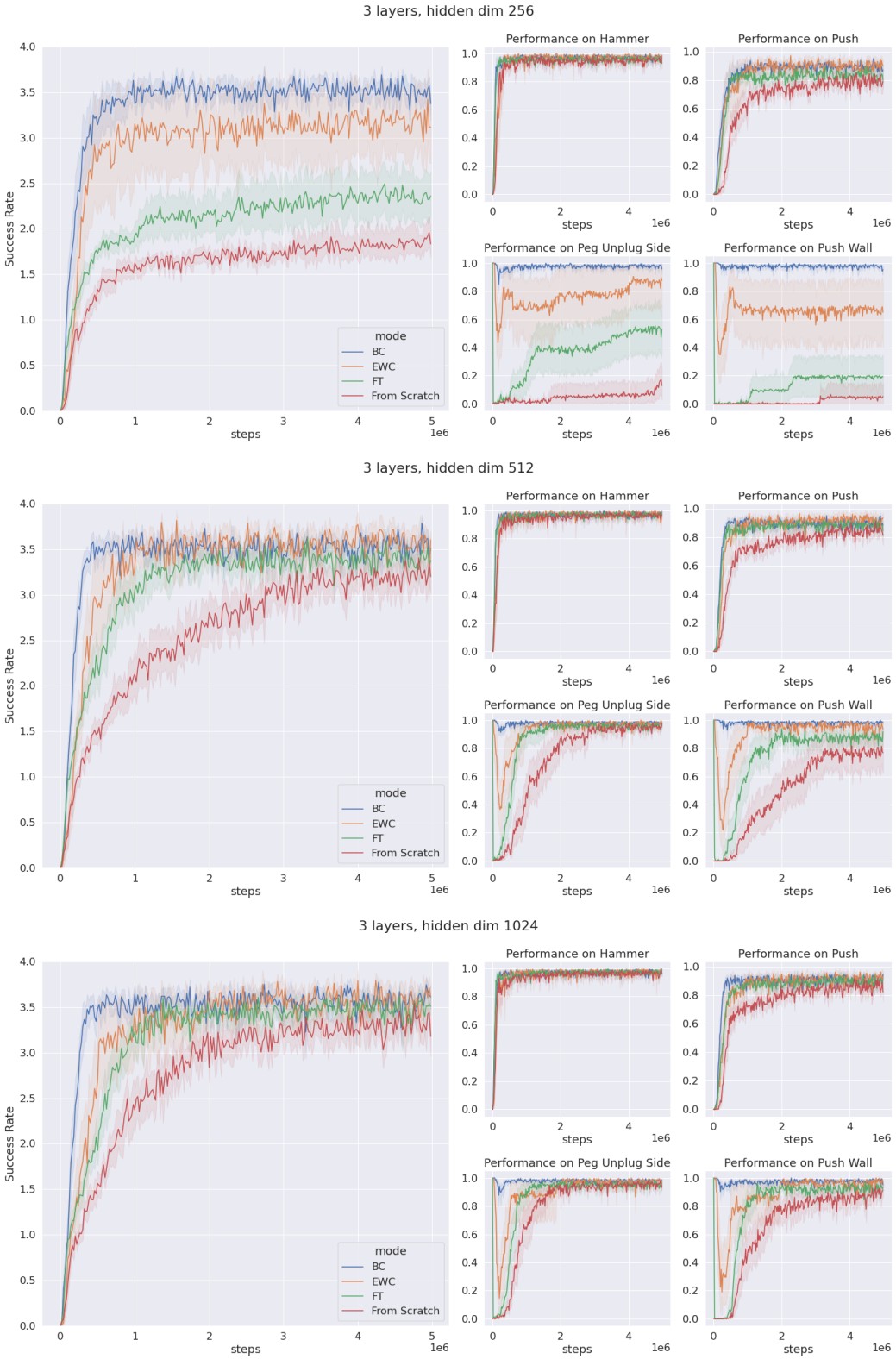

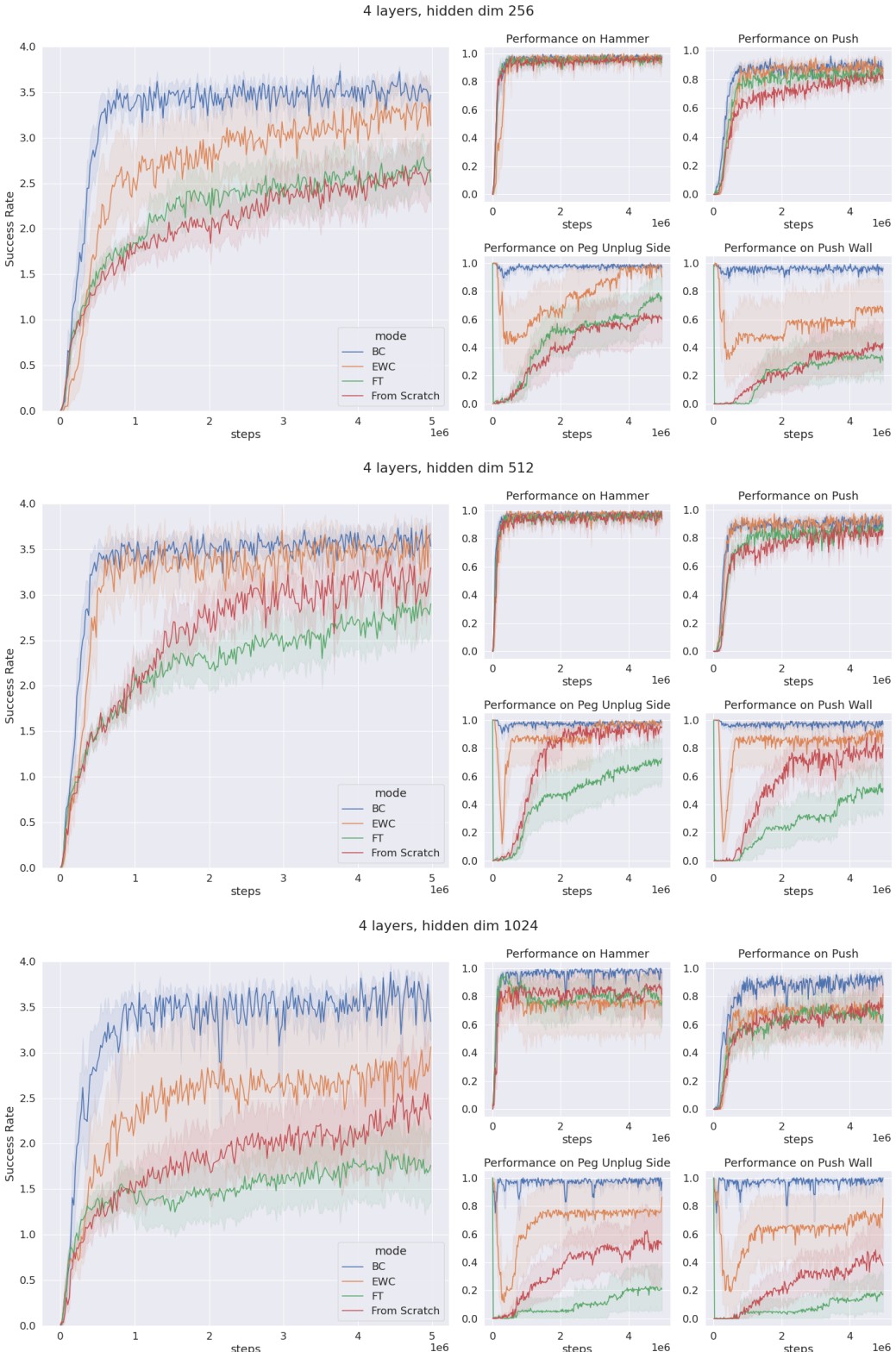

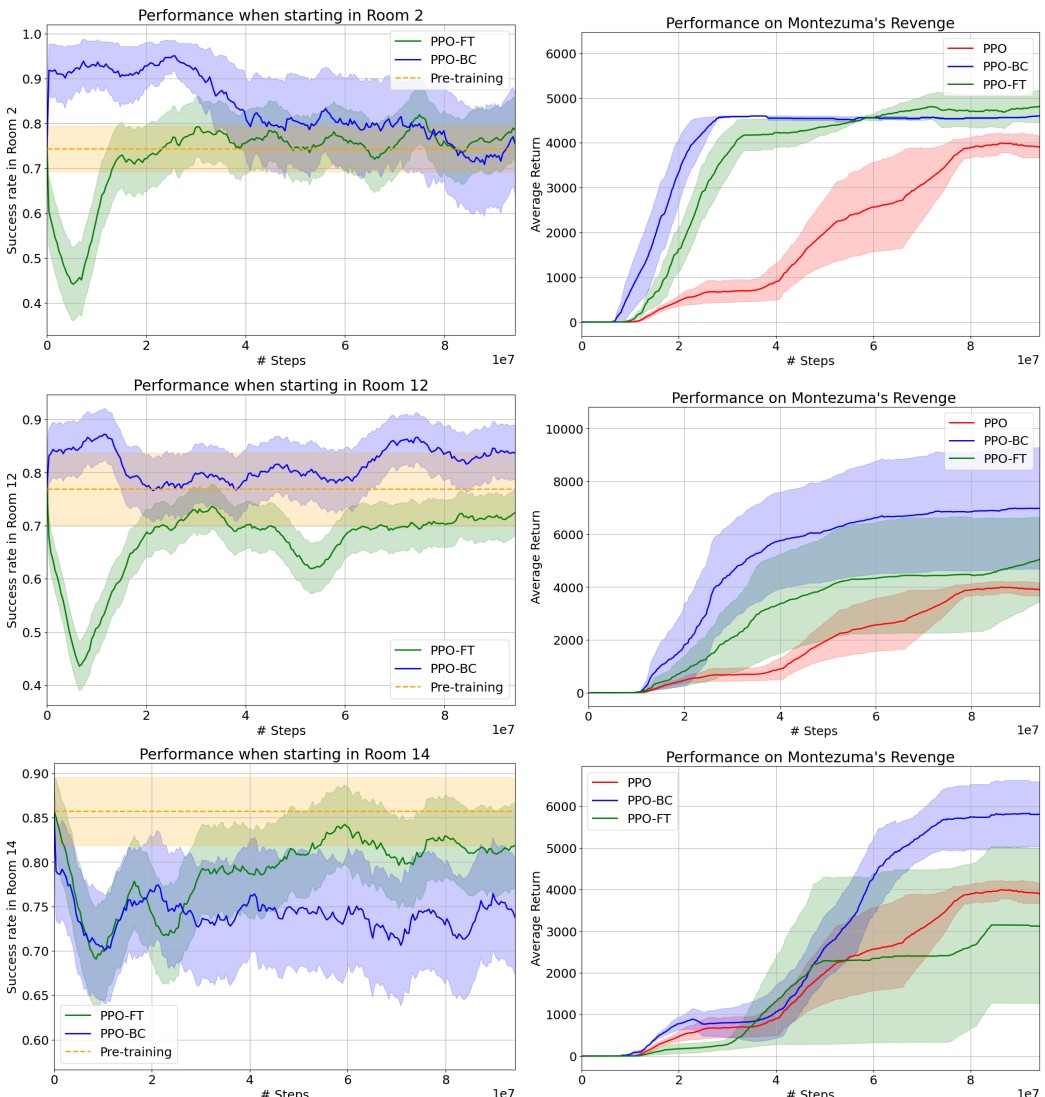

(a) Success rate in rooms during fine-tuning when initialized in that room.

(b) Average return throughout the training. PPO-FT and PPO-BC are fine-tuned, while PPO is learned from scratch.

Figure 19: State coverage gap in Montezuma's Revenge.

# E   ADDITIONAL MONTEZUMA'S REVENGE RESULTS

**Analysis of forgetting with different pre-training schemes**   We perform additional experiments on three different rooms analogous to one from Section 3.2. In particular, we are interested in the behaviour of the pre-trained model from a specific room while fine-tuned. Figure 19 shows a significant drop in performance for fine-tuned models without additional forgetting mitigation methods (PPO-FT) just after fine-tuning starts. In contrast, PPO-BC mitigates this effect except for room 14. For all pre-training types, PPO-BC outperforms PPO-FT with respect to the score.

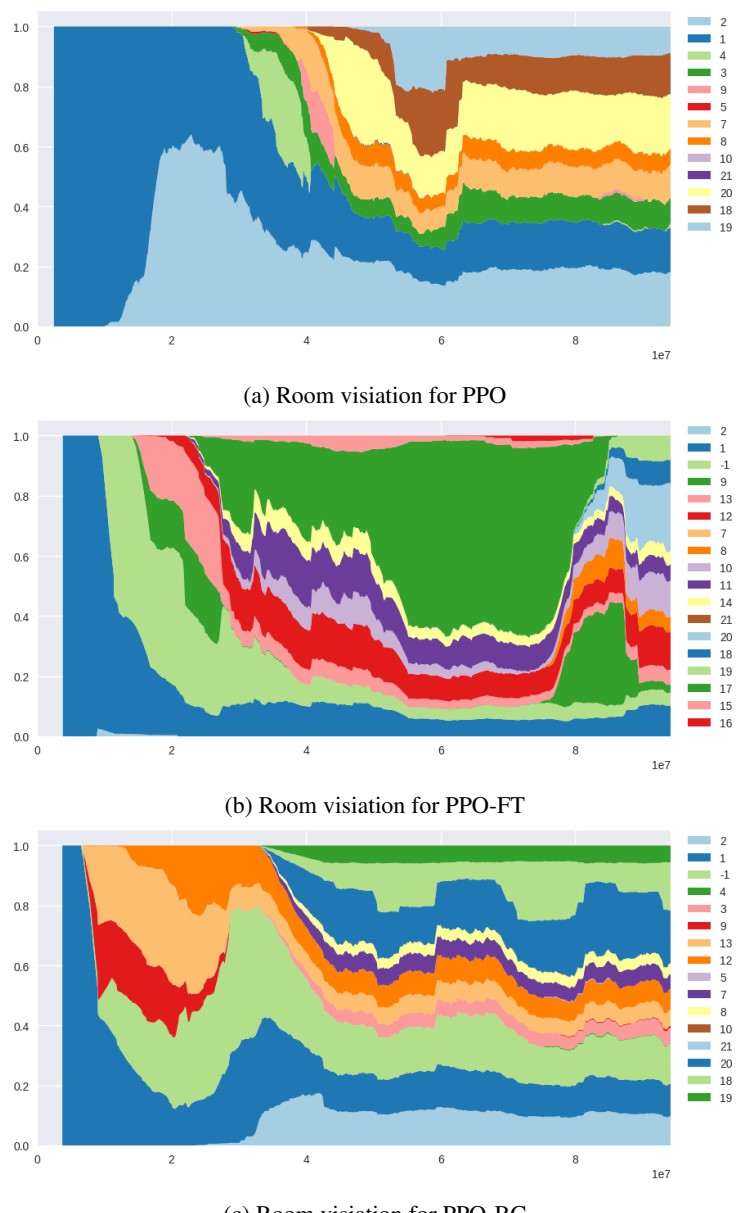

(a) Room visiation for PPO

(b) Room visiation for PPO-FT

(c) Room visiation for PPO-BC

Figure 20: Time spent in different rooms across training for PPO (top), PPO-FT (middle), and PPO-BC (bottom). The agent trained from scratch struggles to explore rooms at the beginning of the training and eventually visits fewer of them than fine-tuned agents.

**Room visitation analysis** In Figure 20, we show how the time spent in different rooms changes across the training for an agent trained from scratch (PPO), fine-tuned agent (PPO-FT), and fine-tuned agent with KL loss (PPO-BC). The agent trained from scratch spends a significant amount of time learning to escape the first two rooms, while later, the agent can navigate between different rooms more effectively. This tendency is not visible for fine-tuned agents, which relatively easily escape the first room. It is important to note that the agent visits only a part of all possible rooms (13 out of a total of 23), and some rooms are only visited a few times across training. Most likely, the reason behind that might be insufficient intrinsic motivation from the RND component and navigation towards rooms where the agent already found the reward.

# F   ADDITIONAL NETHACK RESULTS

In this section, we provide additional results for the NetHack experiments shown in Section 4.

**Different approaches to distillation**   In our main NetHack experiments, we use the following behavioral cloning loss:

$$\mathcal{L}_{BC}(\theta) = \mathbb{E}_{s \sim \mathcal{B}}[D_{KL}^s(\pi_*(\cdot|s) \parallel \pi_\theta(\cdot|s))], \qquad (5)$$

where $\mathcal{B}$ is a buffer of data gathered by the AutoAscend, and $\pi_*$ is the pre-trained policy. However, there are more possible ways to apply distillation to this problem. Here, we study two other approaches – **Kickstarting** and **Behavioral Cloning with Expert Labels**.

In **Kickstarting** (KS) Schmitt et al. (2018), we use a very similar loss, but now we apply KL on the data gathered online by the student. More formally:

$$\mathcal{L}_{KS}(\theta) = \mathbb{E}_{s \sim \mathcal{B}_\theta}[D_{KL}^s(\pi_*(\cdot|s) \parallel \pi_\theta(\cdot|s))], \qquad (6)$$

where $\mathcal{B}_\theta$ denotes a buffer of data gathered by the online policy $\pi_\theta$.

In **Behavioral Cloning with Expert Label** (BCEL), we use the states gathered by the expert, same as in behavioral cloning, but we use a different target. Namely, we use the prediction from the AutoAscend dataset on which we pre-trained $\pi^*$. Formally,

$$\mathcal{L}_{BCEL}(\theta) = \mathbb{E}_{s \sim \mathcal{B}}[D_{KL}^s(y^* \parallel \pi_\theta(\cdot|s))], \qquad (7)$$

where $y^*$ is the label from the AutoAscend dataset corresponding to the action taken by the expert agent. As such, there is a subtle difference between $L_{BC}$ and $L_{BCEL}$. In the first one, we attempt to preserve the knowledge of $\pi^*$, in the second one we try to optimize the same loss that was used to train $\pi^*$. If $\pi^*$ were to perfectly clone the AutoAscend dataset, these two approaches would be exactly the same, but in practice, we see that there is a significant discrepancy between these two.

We compare the three approaches and show the results in Figure 21 and Figure 22. First of all, we observe that APPO-KS performs worse than the other types of distillation. We attribute this gap to the fact that APPO-KS applies regularization on the online data where the pre-trained policy performs poorly. The regularization stops the model from learning a better behavior on these states and as such leads to stagnation. On the other hand, quite surprisingly, APPO-BCEL performs slightly worse than APPO-BC, even though it uses targets from the AutoAscend dataset that performs much better than $\pi^*$. We hypothesize that it is simpler for the model to maintain the initial performance with APPO-BC rather than to learn something new with APPO-BCEL.

These results show that one should carefully choose how to apply distillation to the model. At the same time, we do not claim that these rankings will transfer to other settings and that APPO-BC is inherently better than APPO-BCEL and APPO-KS. We believe that one could fix APPO-KS by e.g. decaying the regularization term.

**Forgetting in the fine-tuned models**   In Figure 25, we measure how different methods forget throughout the fine-tuning process. In particular, we measure the KL divergence on the data generated by the teacher (the pre-trained model). While APPO-FT forgets completely, APPO-BC keeps the divergence at fairly low levels.

**Descent statistics**   Although the obtained return is a valuable metric of the performance of a policy, it is far from the only one. Here, for each of the policies introduced in our experiments, we check the performance in terms of the number of levels visited by each policy. The results are shown in Figure 26.

**Sokoban results and discussion**   Here, we discuss the Sokoban levels in NetHack and their unique complexity. n the Sokoban levels, the player has to face monsters and essentially play the game as normal in addition to moving boulders. There are specific unlucky scenarios when the monster

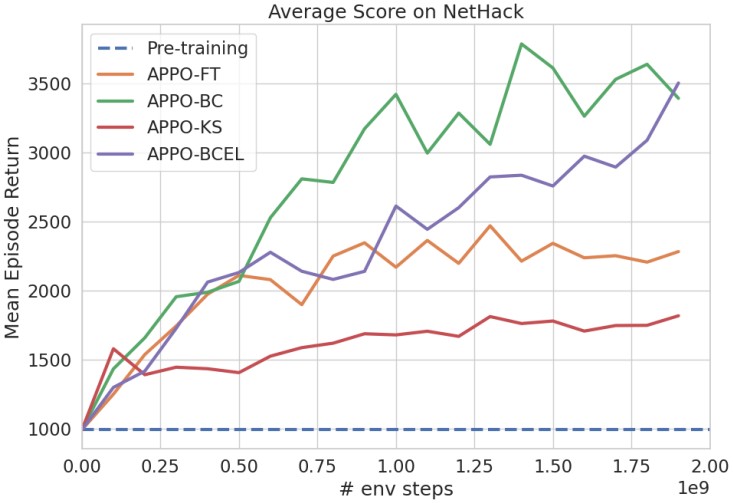

Figure 21: Average return when fine-tuning on NetHack with additional distillation approaches.

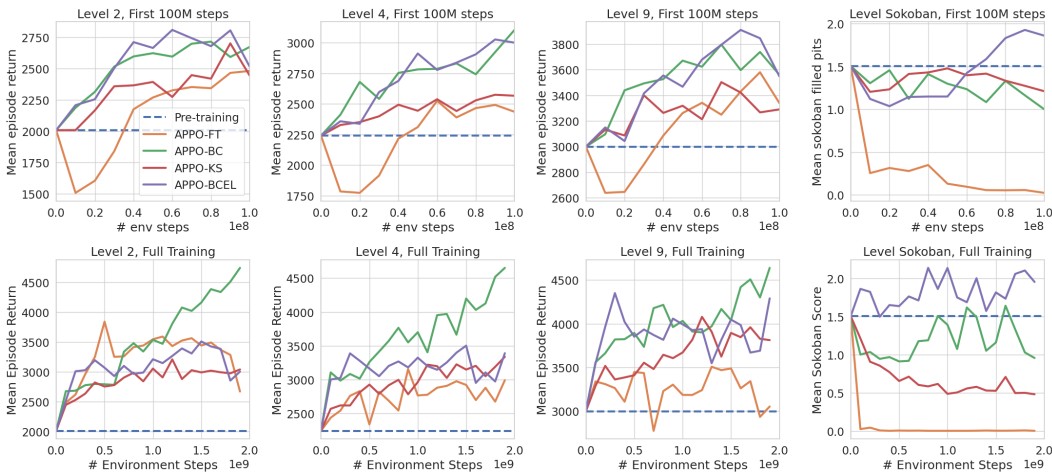

Figure 22: Average return of each of each NetHack method throughout the training with additional approaches to distillation, evaluated when started on a specific dungeon level. The top row shows the initial phase of the training (up to 100M steps), and the bottom row shows the full training (up to 2B steps, note the x-axis scale change). While the distillation approaches showed similar performance in the early stages (top row), APPO-BC stands out in the overall training, particularly on Level 2 and Level 4.

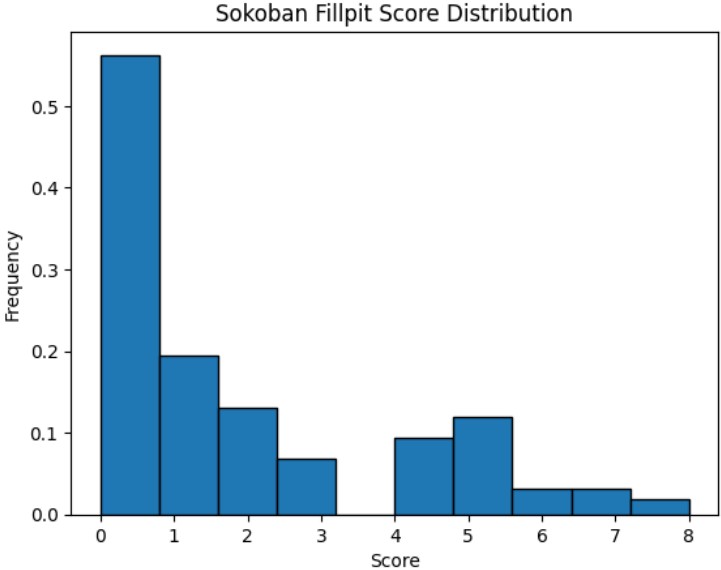

Figure 23: The distribution of how many pits the agent manages to fill in the Sokoban levels.

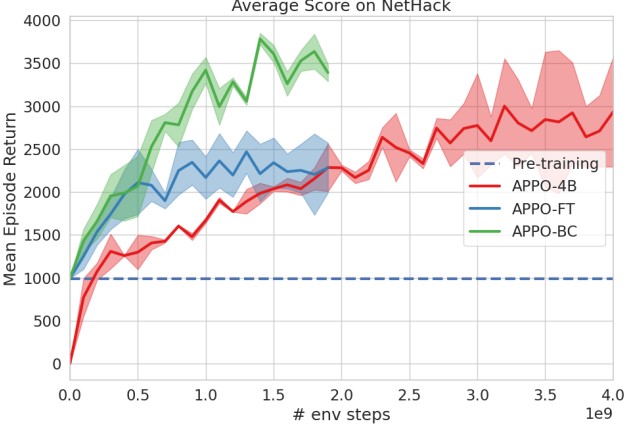

Figure 24: Training from scratch on NetHack, ran with twice the budget of other methods. Although it manages to outperform APPO-FT, APPO-BC still achieves a better results, even with the smaller timestep budget.

will spawn between boulder and pit, which require the agent to find non-trivial ways of killing the monster from behind the boulder. Additionally, the Sokoban levels are encountered fairly late in the game and as such might be relatively underrepresented in the data.

As such, the pre-trained agent that clones the actions of an expert policy starts with a score of 1.5 filled pits. It is worth noting that the mean is somewhat skewed due to the number of episodes when the agent chooses to ignore Sokoban completely and go into a different branch of the dungeon. This is a viable strategy, since the agent might try to increase the score in a different manner. In Figure 23, we show a histogram of the number of pits filled by the pre-trained agent. It is worth highlighting that it does not progress at the problem in over 50% of trajectories, but in over 25% percent of trajectories it solves at least half of the level.

**Training from scratch with more steps** Although in our experiments we observe that fine-tuning with knowledge distillation methods significantly outperforms training from scratch with a given

budget of timesteps, it is not clear if a model trained from scratch can finally catch up given enough timesteps. To investigate this, we run APPO with 4B steps rather than 2B as in the previous experiments. In Figure 24, we compare it with pure fine-tuning. We observe that although APPO manages to finally overtake pure fine-tuning, in the end it still falls short of knowledge retention-based APPO-BC that achieved score of over 3500.

**Full results** In Figure 27 we show the full results of our fine-tuning experiments with evaluation on specific levels. In addition to the standard average return metrics, we provide additional information such as the score wrt. to other objectives defined in NLE Küttler et al. (2020) (e.g. eating food, gathering gold, descending through the dungeon) and additional player statistics (experience level, energy).

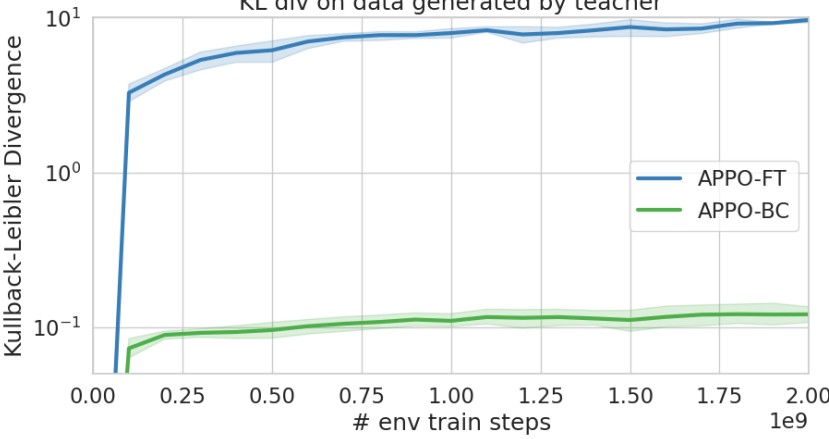

Figure 25: We measure how outputs of the policy change to the pre-trained one during training every 100 million steps, note the log-scale. We can see that APPO-BC aggressively prevents changes to the policy.

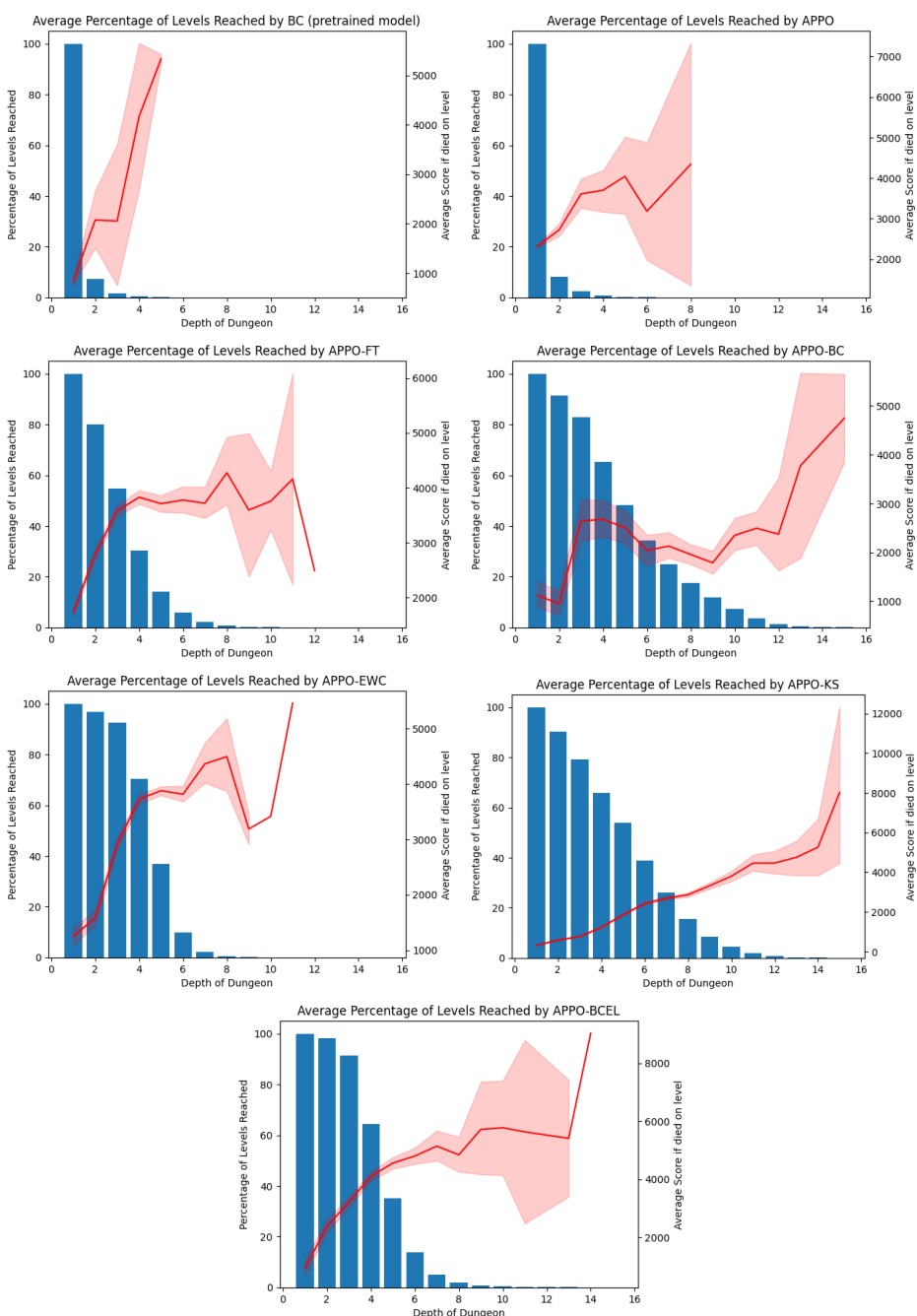

Figure 26: Average percentage of levels reached by different methods. The red line shows the average return obtained in trajectories that finished on this level, while the blue bar shows the percentage of trajectories that reached a given level.

Figure 27: NetHack performance on various metrics when starting from specific levels

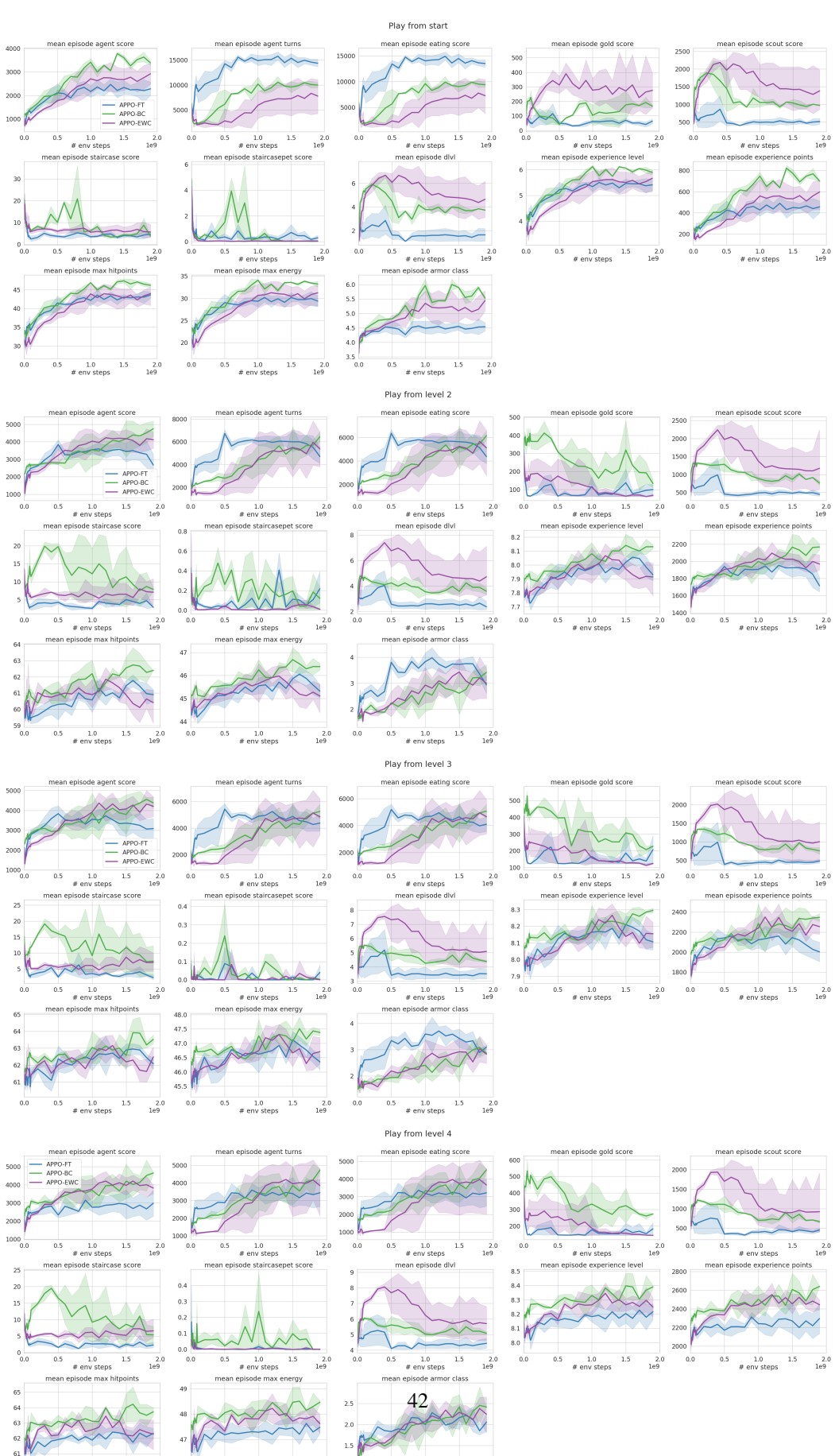

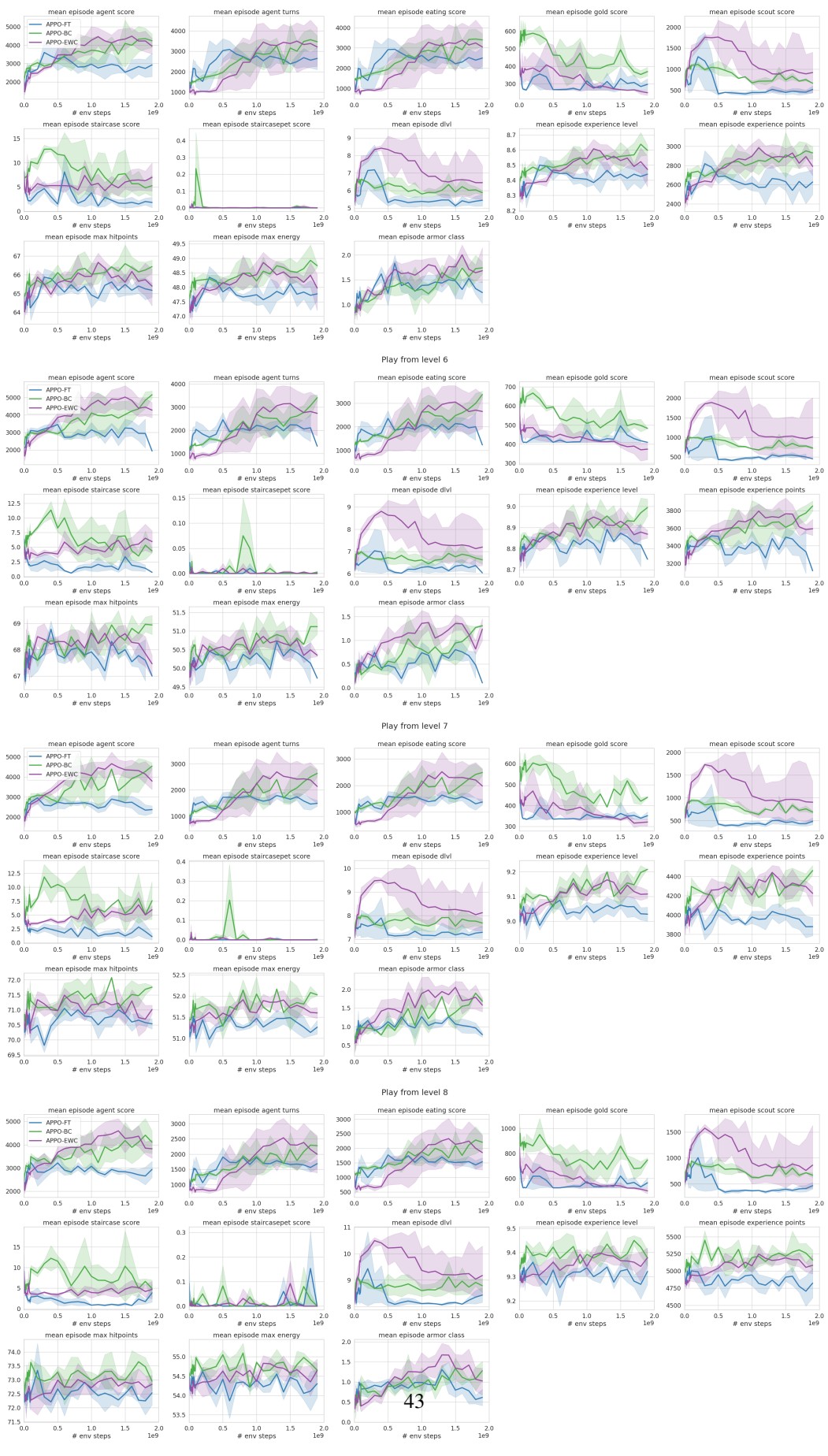

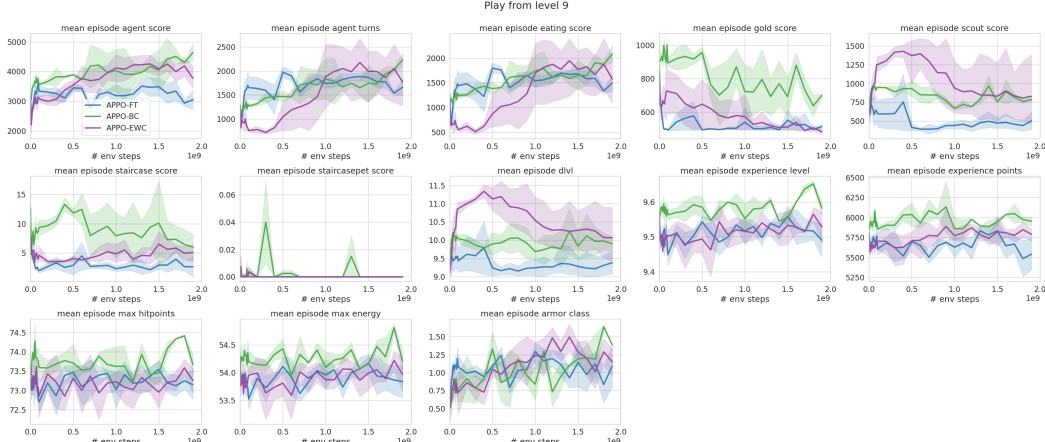

