# OpenReview forum: "The Role of Forgetting in Fine-Tuning Reinforcement Learning Models"
_ICLR.cc/2024/Conference — Submitted to ICLR 2024_

### Official Review · Reviewer_5ZRC · 2023-10-30

**Soundness:** 3 good
**Presentation:** 3 good
**Contribution:** 3 good
**Rating:** 8
**Confidence:** 4

**Summary:**

This work investigates catastrophic forgetting in fine-tuning pre-trained reinforcement learning (RL) policies on subsequent tasks sequentially in a stationary environment while data distribution shifts. It first shows how fine-tuned policies would deteriorate in performance for previous tasks. Then, the paper identifies two conditions in which forgetting occurs, namely, state coverage gap and imperfect cloning gap. Experimentally, the work further shows how existing knowledge retention methods like elastic weight consolidation (EWC) mitigate forgetting during the fine-tuning process.

**Strengths:**

1. This is an important research problem for both the understanding of deep RL training and potential practical deployments. We have seen extensive studies on fine-tuning of supervised learning. The same aspect in RL is relatively less studied. As deep RL moves towards large-scale pretraining, understanding the best practices of fine-tuning with downstream tasks is crucial.
2. The paper shows strong empirical analysis in understanding the problem, accompanied with extensive experimental results. I find the identification of the two conditions to be informative to researchers of this subfield
3. The paper in general is clearly written with key results elaborately explained.
4. Experimental results are comprehensively displayed. I particularly find figure 4 to be intuitive and helpful in visualizing the forgetting phenomenon.

**Weaknesses:**

1. The choice of benchmarking algorithms for knowledge retention, although somewhat representative of existing methods, does not quite match with state-of-the-art approaches. Newer methods like [1], if added, can strengthen the conclusions of the paper.
2. It is unclear to me how is this setting different from continual/lifelong RL
3. [Minor] the term ‘realistic RL algorithms’ is confusing

[1] Ben-Iwhiwhu, E., Nath, S., Pilly, P. K., Kolouri, S., & Soltoggio, A. (2022). Lifelong reinforcement learning with modulating masks. arXiv preprint arXiv:2212.11110.

**Questions:**

1. How is non-stationary enironment different from data shifts in stationary environment? Is it not the same underlying data shift problem?
2. What if we pretrain ‘CLOSE’ states first instead? Do we see better forward transfer?
3. Can the authors provide their views on why pre-trained models (counterintuitively) do not seem to exhibit any signs of positive transfer? Existing methods do seem insufficient for RL to leverage pretraining
4. Why is EWC missing in some of the subsequent experiments?

---

> ### Author Response · Authors · 2023-11-15
> **Response to Reviewer 5ZRC**
>
> We thank the Reviewer for the kind and insightful comments about our work. During the rebuttal period we performed additional experiments and improved the manuscript. In the general answer we list introduced changes and provide a clarification of the differences for the CL setup. We hope it is sufficient, please let us know if you have any more questions or comments.
>
> > How is a non-stationary environment different from data shifts in a stationary environment? Is it not the same underlying data shift problem?
>
> On a high level view, the problem which we study can be framed as a ‘data shift’ problem. The difference from the standard online RL setup is the ‘direction of changes’ the training experiences FAR states only after seeing CLOSE ones. In the pre-training/fine-tuning scenario considered in this work, it might be otherwise, and, as we argue (see the General response), this will be more likely as researchers progress with building foundational models in RL. In this work, we highlight that this revered order may cause challenges. We hope this answers your question, please let us know, if you have any suggestions or further questions. This bit is quite challenging to describe in an unambiguous way.
>
>
> > Newer methods like [1], if added, can strengthen the conclusions of the paper.
> > Why is EWC missing in some of the subsequent experiments?
>
> Since our main goal is to show and analyze the forgetting problem in fine-tuning RL models, we decided to use relatively simple methods to avoid multiple overlapping effects. We discovered behavioral cloning to be better and more robust than L2 or EWC, so we decided to use it. For the same reason, we do not check more sophisticated continual learning methods on our benchmark. We believe that incorporating more sophisticated CL methods to the fine-tuning problem is an important next step, which we now mention in the Limitations section.
>
> > What if we pretrain ‘CLOSE’ states first instead? Do we see better forward transfer?
>
> We thank the Reviewer for this interesting ablation study. To test it, we run additional experiments in the robotic domain, see Appendix D, paragraph “Other sequences”. In short, we do not observe forgetting and there is no significant gap between vanilla fine-tuning and knowledge retention methods. This is further evidence that the forgetting caused by the state visitation gap is at the heart of the problem of transfer.
>
> > Can the authors provide their views on why pre-trained models (counterintuitively) do not seem to exhibit any signs of positive transfer? Existing methods do seem insufficient for RL to leverage pretraining
>
> Pre-trained models exhibit some positive transfer, which, however, varies substantially unless forgetting is under control. Our work is one of the first initial steps towards understanding this issue and building its taxonomy (see also [2] for an interesting benchmark of transfer in CL). Additionally, in the experiment suggested above by the Reviewer, we show that without the state visitation gap we can see a significant transfer even without any knowledge retention methods. At the same time, we do agree with the Reviewer that further research is needed in order to use pre-trained knowledge more efficiently.
>
>
> > [Minor] the term ‘realistic RL algorithms’ is confusing
>
> We agree that the term is not precise and we get rid of it in the revised version of the paper. Thanks for pointing this out.
>
> [1] Ben-Iwhiwhu, E., Nath, S., Pilly, P. K., Kolouri, S., & Soltoggio, A. (2022). Lifelong reinforcement learning with modulating masks. arXiv preprint arXiv:2212.11110. \
> [2] Wolczyk et al, Disentangling Transfer in Continual Reinforcement Learning, NeurIPS’22.

---

> ### Author Response · Authors · 2023-11-20
> **Response to Reviewer 5ZRC30, revision 2**
>
> Following your suggestion, we have conducted additional experiments incorporating EWC in Montezuma’s Revenge. While we observed that EWC provides some benefit in mitigating forgetting, we found that behavioral cloning remains a more effective and robust approach for addressing the forgetting problem in our specific context. The results and a detailed analysis of these experiments are now included in our revised manuscript (figure 5, 6 and 7). We believe this comparison adds valuable insights to the discussion of different methods for addressing forgetting in fine-tuning RL models.

---

> > ### Comment · Reviewer_5ZRC · 2023-11-21
> >
> > Thank you very much for addressing my questions together with the additional experiments and discussion. My rating of this work remains to be positive. It is an informative and detailed contribution to this research area.

---

### Official Review · Reviewer_bshN · 2023-10-31

**Soundness:** 2 fair
**Presentation:** 3 good
**Contribution:** 1 poor
**Rating:** 3
**Confidence:** 4

**Summary:**

This is an experimental paper that studies the forgetting issue in finetuning pre-trained models with RL. The paper focuses on two special cases of the problem: state coverage gap and imperfect cloning gap. To study the two problems respectively, the paper compares several existing methods in Meta-World, Montezuma's Revenge, and NetHack. Results shows that RL with behavior cloning on the pre-training dataset outperforms other methods, maintaining the pre-trained capabilities better during RL.

**Strengths:**

1. Forgetting of the previously learned skills is a problem worth studying in RL.

2. The paper refines this problem into two cases and conducts appropriate experimental evaluation.

**Weaknesses:**

1. As an experimental paper studying forgetting, it lacks evaluation on many related methods. The paper only evaluates two kinds of  methods: parameter regularization and behavior cloning. But there exists many other methods addressing the forgetting issue in the literature of continual RL and finetuning with RL, like using offline RL over previous data [1], adding KL-divergence loss to the pre-trained policy on the online data [2], and a lot of methods in sharing representations and structures [3].

2. The experimental results do not provide different insights into the two problems. All the results demonstrate that Finetuning+BC outperforms other methods and the vanilla Finetuning method suffers from forgetting on the FAR states. But beyond that, the results lack further analysis of the two problems and do not reflect the significance of dividing the forgetting problem into the two types.

3. The paper has no novel contributions in methods and techniques. It also cannot provide insights in how to better address forgetting in the future work.

[1] Modular Lifelong Reinforcement Learning via Neural Composition.

[2] Video PreTraining (VPT): Learning to Act by Watching Unlabeled Online Videos.

[3] Towards continual reinforcement learning: A review and perspectives.

**Questions:**

1. Can the experimental results provide different insights into the two problems? In addition to these two problems, does the problem of forgetting include other cases?

2. Based on the experimental results, are there any insights in improving the existing methods or further addressing the forgetting problem?

---

> ### Author Response · Authors · 2023-11-15
> **Response to Reviewer bshN**
>
> We thank the Reviewer for their insightful review and useful comments. We note that in the revised version we provide multiple clarifications and new experiments, listed in the general answer. We would be happy to address more questions and suggestions.
>
> > As an experimental paper studying forgetting, it lacks evaluation on many related methods
>
> We thank the reviewer for this suggestion. Since our main goal is to show and analyze the forgetting problem in fine-tuning RL models, as we discuss in General response, we decided to use relatively simple tools to mitigate it. We discovered behavioral cloning to be a better performing and a more robust approach than L2 or EWC, so we decided to use it. For the same reason, we do not check more sophisticated continual learning methods on our benchmark, even though we believe they might also lead to amazing results. However, we agree with the Reviewer that testing other approaches would be a great idea for further studies.
>
> We thank the reviewer for providing valuable references. Authors of [1] use batch RL in order to maintain performance on the previous tasks. We take advantage of the fact that SAC is an off-policy method that has been a basis for multiple offline RL algorithms [4, 5] and we are currently running experiments where we keep the data from the previous tasks in the buffer. We will update the experiments once they are done. At the same time, we would like to point out that this approach would not be viable in the case of NetHack, where we pre-train the model using behavioral cloning and as such we do not have a critic network to clone.
>
> VPT [2] includes a different type of distillation, but they do not specify what data exactly they are using, e.g. are the trajectories samples from the student or the teacher policy. In order to better understand this problem, we perform further studies of two variations of distillation: (a) where the trajectories are obtained from the student (i.e. known as kickstarting in the literature [6]) (b) and where the targets are obtained from the expert dataset that the pre-trained policy aimed to clone rather than from the pre-trained policy itself. Surprisingly, we find that both of these approaches perform worse than our standard strategy. The full results and analysis are shown in Appendix F, paragraph “Different approaches to distillation”.
>
> Finally, although we believe methods relying on sharing representations and structures [3] could bring substantial improvements to the studied problem, to the best of our knowledge most approaches in this family require clear task separation. In the RL fine-tuning setting it might not be trivial to partition the state space into FAR and CLOSE states, and as such applying these methods would be difficult. However, we believe finding efficient ways to apply such approaches to be an important future research direction. We agree with the reviewer that incorporating a wider variety of CL methods to the fine-tuning problem would be useful, and we add this point to the Limitations section in the revised manuscript.
>
> > Can the experimental results provide different insights into the two problems?
>
> We use the two problems to highlight natural circumstances in which forgetting of pre-trained capabilities occurs. That is, when the state spaces of the upstream and downstream task overlap but are not identical [7, 8, 9], and when the model is pretrained on offline data and finetuned online [10, 11, 12].
>
> There are subtle differences between these two settings. In particular, with the imperfect cloning gap, it is far more difficult to assess in which states we should minimize forgetting. As such, one should be careful about where to apply distillation techniques, as we show through new experiments in Appendix F.
>
> > In addition to these two problems, does the problem of forgetting include other cases?
>
> Another natural case is the related set of tasks that share the same state space but differ in the reward functions.
>
> > Based on the experimental results, are there any insights in improving the existing methods or further addressing the forgetting problem?
>
> An important conclusion (see Appendix F) is that it is probably beneficial to use regularization techniques which ‘strength’ depends on the state and the phase of training. Yes, in the future work we provide multiple open research directions.
>
> [4] Conservative q-learning for offline reinforcement learning \
> [5] Uncertainty-Based Offline Reinforcement Learning with Diversified Q-Ensemble \
> [6] Kickstarting Deep Reinforcement Learning \
> [7] Probing Transfer in Deep Reinforcement Learning without Task Engineering \
> [8] Progressive neural networks. \
> [9] Actor-mimic: Deep multitask and transfer reinforcement learning \
> [10] Video PreTraining (VPT): Learning to Act by Watching Unlabeled Online Videos \
> [11] Accelerating Online Reinforcement Learning with Offline Datasets \
> [12] Adaptive Policy Learning for Offline-to-Online Reinforcement Learning \

---

> ### Author Response · Authors · 2023-11-20
> **Response to Reviewer bshN31, revision 2**
>
> Following your suggestion, we implemented and evaluated an additional method on Continual World; Episodic Memory (EM), in which we populate SAC’s replay buffer with trajectories from the pre-training tasks. We observe that EM works slightly better than EWC, but worse than behavioral cloning.
>
> Additionally, we run experiments with EWC on Montezuma’s Revenge. The results show that EWC provides improvements, but is not as good as Behavioral Cloning. Details of the experiments are presented in Figure 5.
>
> We believe that overall new experiments and analyses considerably strengthen our paper. We, thus, gently ask you to consider raising the score or pointing out remaining deficiencies.

---

### Official Review · Reviewer_BuFP · 2023-11-04

**Soundness:** 3 good
**Presentation:** 3 good
**Contribution:** 1 poor
**Rating:** 3
**Confidence:** 4

**Summary:**

Summary:

This paper studies fine-tuning in RL, and specifically the issue of forgetting and potential mitigation strategies. They demonstrate in several settings (simulated robot manipulation, Montezuma's Revenge and NetHack) that if a policy is pretrained on some part of the state space which is far from the initial state distribution, the knowledge is often forgotten and there are little to no improvements over training from scratch. They furthermore investigate different knowledge retention strategies (such as L2 penalties between pretrained and fine-tuned policy weights, possibly weighted by fisher information, as well as simple BC regularization on the pretraining data). They find that BC regularization helps the most, and can help prevent forgetting the behaviors encoded in the pretrained policy.

**Strengths:**

- The paper's main takeaway message, that adding BC regularization helps avoid forgetting previous behaviors during fine-tuning, is well supported by the experiments. This is demonstrated in 3 environment, including continuous control (MetaWorld), a pixel-based Atari game (Montezuma's Revenge) and a procedurally generated, long-horizon game with complex dynamics (NetHack).

- The paper does a nice job with their analysis and visualizations illustrating the forgetting behavior.

**Weaknesses:**

- The main takeaway, which is essentially that co-training on the old tasks prevents forgetting when learning a new task, is pretty unsurprising and has been demonstrated before in previous works in continual learning both for the supervised case and the RL case. It's not clear what the contribution of this work adds.
- An obvious downside of co-training on previous tasks is that the memory requirement increases linearly with the number of tasks and the computation increases quadratically - this is not adequately discussed.
- It would have been nice to include result for the L2 and EWC on Montezuma and NetHack.

**Questions:**

Some suggestions on the writing:

- In the intro, it would be helpful to give a bit more details on the "knowledge retention techniques" used to mitigate the forgetting problems. Currently, the ready does not have much idea on the methodological aspects going into the paper.

- Example in 2-state MDP: the notation here is confusing. Both $theta$ and $f_\theta$ are used before being defined. Please add the definitions in the main text.

---

> ### Author Response · Authors · 2023-11-15
> **Response to Reviewer BuFP**
>
> We would like to thank the Reviewer for the kind comments about the paper’s experimental value, and for the interesting suggestions. We address them below. Additionally, we conducted new experiments, please see the general response for a full list.
>
>
> > The main takeaway, which is essentially that co-training on the old tasks prevents forgetting when learning a new task, is pretty unsurprising and has been demonstrated before in previous works in continual learning both for the supervised case and the RL case. It's not clear what the contribution of this work adds.
>
> We respectfully disagree with the Reviewer regarding the value of the main takeaway. As detailed in the general answer, we consider the conceptualization of 'forgetting of pre-trained capabilities' to be non-obvious and important. We study a new perspective. More precisely, we are not aware of any work comprehensively showing that forgetting is detrimental to transfer in RL fine-tuning scenarios. Please let us know if you know any; we would be happy to include and discuss them.
>
> > An obvious downside of co-training on previous tasks is that the memory requirement increases linearly with the number of tasks and the computation increases quadratically - this is not adequately discussed.
>
> We thank the Reviewer for raising this important issue. To address this we:
> - include additional discussion in the Limitations section,
> - provide additional experiments in Appendix D, which indicate that as little as 100 samples is sufficient to observe a significant improvement over fine-tuning. We typically use 10000 transitions sampled out of 5M steps for training, which amounts to roughly 3.5MB of memory. We argue that, compared to the current hardware capabilities, this is negligible.
>
> > It would have been nice to include results for the L2 and EWC on Montezuma and NetHack.
>
> In the robotic experiments, we discovered that behavioral cloning performs better and is more robust than L2 or EWC, and we anticipated the same for the other environments. However, we consider designing and benchmarking other methods to be a very important direction of future work, which is indicated in the Limitations section.
>
> > In the intro, it would be helpful to give a bit more details on the "knowledge retention techniques”
> > Example in 2-state MDP: the notation here is confusing.
>
> We thank the Reviewer for these suggestions. In the revised version of the manuscript we added a short introduction to knowledge retention techniques and we clarified the notation in the MDP examples. Please let us know if you have any further comments.

---

> ### Author Response · Authors · 2023-11-20
> **Response to Reviewer BuFP04, revision 2**
>
> We are happy to announce new results.
>
> Following your suggestion, we ran additional experiments with EWC on Montezuma’s Revenge. Our findings indicate that while EWC provides improvements, it still fails short compared to performance achieved by Behavioral Cloning. Detailed results and analyses of these experiments are presented in Figure 5 the revised manuscript.
>
> Thank you once again for your constructive feedback, which improves our work. We believe that our paper improved considerably during the rebuttal period. We, thus, gently ask you to consider raising the score, or providing us with further questions.

---

### Official Review · Reviewer_dWac · 2023-11-05

**Soundness:** 3 good
**Presentation:** 3 good
**Contribution:** 2 fair
**Rating:** 6
**Confidence:** 4

**Summary:**

This paper examines finetuning of pretrained RL agents in a single environment. Two problematic mechanisms are identified. A state coverage gap occurs when the agent is pretrained on a part of the state space but, in the fintetuning phase, has to first learn a policy on a different part. Then, the policy on the first part of the state space is lost during finetuning and must be relearned. The second, the imperfect cloning gap, occurs when the agent is pretrained through imitation learning. As the policy is finetuned, the performance on states later in trajectories also degrades.
The use of behavioru cloning on states from the first task and other forgetting mitigation techniques are shown to solve these issues. A variety of environments are considered including toy tasks, metaworld and Nethack to demonstrate the problem and the utility of the solutions.

**Strengths:**

- There are extensive experiments on a variety of environments. The sequence of metaworld tasks was an interesting custom addition.

- The identified problem could be relevant in a variety of practical settings. The imperfect cloning gap seems to be particularly applicable since we may often want to start with imitation learning from previous policies if possible.

- There was sufficient detail in the text to understand the experiments and the figures were clear in general.

**Weaknesses:**

- The clarity of certain sections could be improved with more details. For example, for the initial toy example, it would help if the main text explained the motivation behind the MDPs design a little more: why that particular choice of transitions and rewards was made. Also, $f_0$ is not described in the main text and it looks like subfigures b) and c) are interchanged for this example.

- The main proposed solution, behaviour cloning from the pretrained policy, seems to be somewhat limited. Behaviour cloning is inherently limited by the quality of the pretrained policy. The experiments show that it's possible to retain the pretrained policy's performance but not exceed it.

- While novelty is difficult to judge, it seems like the identified problematic phenomena are facets of catastrophic forgetting i.e. the idea that neural networks will forget on certain parts of the input space after being trained on others. In this view, it's not too surprising that pretraining on one part of the state space will lead to a detoriation of performance in another. I can appreciate that there's value in demonstrating this in an RL setting though.

**Questions:**

- Are the benefits of pretraining purely from learning a good policy? Are there benefits due to the representations learned in the pretraining phase?

- Have you experimented with the agent only learning a decent, but not great, policy on the FAR tasks? Could we expect to surpass the performance of the pretrained policy? Using behaviour cloning would seem to be limited by the pretrained policy.

- Have you considered off-policy methods? It seems like there may be an advantage since these methods could simply keep around samples (or trajectories) from previous tasks in the replay buffer to learn from without having to necessarily imitate the previous behaviours.

- In the robotic sequence task, have you tried pretraining on the second and third tasks? Are there any learning benefits for the 4th task if you do so?

- In Montezuma's revenge, how far is the agent able to reach without pretraining? Does it get past room 7 consistently? It would be nice to see the overall learning curves of the agent that has been pretrained vs. the one that has not.

- Nethack levels are generated procedurally. How are the sequence of levels chosen for these experiments? When the agent is reinitialized, is it to a fixed level with the same seed?

- For the Nethack experiment, Fig.6, how come the learning curve for finetuning only matches that of the original agent after 2 billion steps? It looks like, if training continued further, the pretrained agent would even do worse.

- The Sokoban results (fig.7) seem to be fairly poor for both agents since they can only fill less than 1.5 pits on average---not close to a solution. Is this to be expected?

Minor points:
- I would consider moving some more of the results from the appendix to the main text since it looks like there's still space remaining.

- In Fig. 4, I would consider changing the text "pre-trained optimal policy" to "pre-trained expert policy" since we don't necessarily have the optimal policy in those environments.

---

> ### Author Response · Authors · 2023-11-15
> **Response to Reviewer dWac**
>
> We thank the Reviewer for a meticulous review and the many useful suggestions. We are grateful for them and we are happy to accept more.
>
> As to the novelty, we indeed focus on the RL setting and we consider the value of our work to lay in the conceptualization of the influence of forgetting on transfer. Please see the general answer for more details. We also note that many new experiments have been added in the revised version.
>
>
> > The clarity of certain sections could be improved with more details.
>
> Thank you for bringing up the issue of clarity of the MDP examples. We have made improvements in the revised version. We also enhanced the clarity of the paper in other sections. All changes are highlighted in blue. Please let us know if you have any more suggestions.
>
> > Are the benefits of pretraining purely from learning a good policy? Are there benefits due to the representations learned in the pretraining phase?
>
> We thank the reviewer for these questions – the issue of disentangling the impact of the policy from the impact of the representation is an interesting one. Although there are several studies on the impact of representation learning on transfer in supervised learning [1, 2], the same question in RL remains relatively understudied. We run new experiments to better understand this problem. Please see Appendix D, paragraph “Impact of representation vs policy on transfer”, where we describe experiments with resetting the last layer of the networks before starting the training. As such, the policy at the beginning is random even on the tasks known from pre-training, but has features relevant to solving the downstream tasks. The results suggest that in fine-tuning the benefits come from transferring both policy and representations, and for EWC in particular the representation transfer is more prominent.
>
>
> > Have you experimented with the agent only learning a decent, but not great, policy on the FAR tasks? Could we expect to surpass the performance of the pretrained policy? Using behaviour cloning would seem to be limited by the pretrained policy.
>
> We agree this is a relevant concern. However, in our experimental results, in many cases fine-tuning with knowledge retention outperforms the level of the pre-trained model.
>
> For example, on NetHack, the average return of a pre-trained model is 1000, which is outperformed by the BC-regularized fine-tuning agent by a factor of 3.5x, with the same pattern holding for deeper levels (aka FAR states); see Figure 6. On Montezuma’s Revenge, we observe that in terms of completing rooms present in pre-training, PPO-BC manages to outperform the pre-trained model in some cases, see Appendix D, Figure 19. In the robotic manipulation tasks, fine-tuning does not outperform the pre-trained policies since the latter are close to optimal.
>
> We speculate that outperforming the pre-trained policy is due to the fact that regularization is a ‘soft bias’, which can be controlled with hyperparameters (e.g., the KL coefficient), expert data, or the policy network size. As such, this bias can be overwritten in the fine-tuning optimization. Although we found it rather easy to tune, there are also rare negative cases (we detail one of them in Appendix F, paragraph “Different approaches to distillation”). How to efficiently control this 'soft bias' throughout training is an interesting research question, and we included it in the Limitations section.
>
> > Have you considered off-policy methods? It seems like there may be an advantage since these methods could simply keep around samples (or trajectories) from previous tasks in the replay buffer to learn from without having to necessarily imitate the previous behaviours.
>
> We agree with the Reviewer that off-policy methods seem well-suited to deal with the forgetting problem. However, on-policy approaches are often better adjusted to settings that require processing a high number of trajectories to facilitate learning: Montezuma's Revenge and NetHack are examples of such environments. As such, we focus on knowledge retention methods that can be used in a wider variety of problems. At the same time, we agree that this is an interesting research direction and we are currently conducting such experiments on the RoboticSequence benchmark with Soft Actor-Critic. We will update the manuscript once they are ready.

---

> ### Author Response · Authors · 2023-11-15
> **Response to Reviewer dWac, pt. 2**
>
> > In the robotic sequence task, have you tried pretraining on the second and third tasks? Are there any learning benefits for the 4th task if you do so?
>
> We appreciate the suggestion for an interesting experiment. We conduct such a study and present the results in Appendix D, paragraph “Other sequences”, i.e. we pre-train the model on the second and third goal from the downstream environment. The relative performance of all methods is the same as in our original ordering, however, we observe that EWC almost matches the score of BC. We note that in terms of asymptotic performance on the 4th task knowledge retention methods perform much better than vanilla fine-tuning, which in turn is better than a model trained from scratch.
>
> > In the Montezuma's Revenge, how far is the agent able to reach without pretraining? Does it get past room 7 consistently? It would be nice to see the overall learning curves of the agent that has been pretrained vs. the one that has not.
>
> The agent is able to reach FAR states without pre-training, however only in the later stages of training. We include the visualization of time spent in different rooms across training time in Appendix E. It takes the agent learned from scratch more than 3x longer to start entering further rooms when compared with the fine-tuned one.
>
> > Nethack levels are generated procedurally. How are the sequence of levels chosen for these experiments? When the agent is reinitialized, is it to a fixed level with the same seed?
>
> In terms of training, we run the experiments on the standard NLE environment without any modifications: the levels in each episode are generated randomly. The FAR states in this scenario are not any specific levels but rather related to challenges in the later parts of the game. In this vein, the dungeon level N in Figure 7 denotes these states, which are encountered after passing the first N-1 level. These are stochastic, in practice, we collect a sample of 200 states. We thank the reviewer for this comment and we modify the manuscript to include a further explanation of these issues in Section 4 as well as Appendix B.2.
>
> > For the Nethack experiment, Fig.6, how come the learning curve for finetuning only matches that of the original agent after 2 billion steps? It looks like, if training continued further, the pretrained agent would even do worse.
>
> Fine-tuning (solid green line) starts with the performance of the pre-trained agent (dashed blue horizontal line). After 2B steps an agent trained from scratch (solid orange line) matches the performance of the fine-tuned agent around the reward value of 2500. Our preliminary results suggest that the performance of these two plateaus at that level. This is consistent with our main takeaway: knowledge transfer from a pre-trained model is greatly hindered due to forgetting of capabilities on the ‘FAR’ states.
>
>
> > The Sokoban results (fig.7) seem to be fairly poor for both agents since they can only fill less than 1.5 pits on average---not close to a solution. Is this to be expected?
>
> Yes, this is expected. The original Sokoban (i.e., moving boulders) itself is a complex game, save the fact that the Nethack agent still needs to fight monsters. In Appendix F, we added a more detailed explanation of this matter.
>
> > In Fig. 4, I would consider changing the text "pre-trained optimal policy" to "pre-trained expert policy" since we don't necessarily have the optimal policy in those environments.
>
> Thank you for the suggestion, we corrected it in the revised version of the manuscript.
>
> [1] Neyshabur, B., Sedghi, H., & Zhang, C. (2020). What is being transferred in transfer learning?. Advances in neural information processing systems, 33, 512-523. \
> [2] Kornblith, S., Chen, T., Lee, H., & Norouzi, M. (2021). Why do better loss functions lead to less transferable features?. Advances in Neural Information Processing Systems, 34, 28648-28662.

---

> ### Author Response · Authors · 2023-11-20
> **Response to Reviewer dWac05, revision 2**
>
> We are happy to announce new results.
>
> Following your suggestion, we implemented and evaluated an additional method on Continual World; Episodic Memory (EM), in which we populate SAC’s replay buffer with trajectories from the pre-training tasks. We observe that EM works slightly better than EWC, but worse than behavioral cloning.
>
> Additionally, we run experiments with EWC on Montezuma’s Revenge. The results show that EWC provides improvements, but is not as good as Behavioral Cloning. Details of the experiments are presented in Figures 5.
>
> Please let us know if these experiments and comments address your concerns. Otherwise we ask you to consider raising the score.

---

> ### Author Response · Authors · 2023-11-21
>
> Additionally, in the latest revision, we include experiments with EWC on NetHack and we run an experiment to answer the Reviewer’s question about what would happen if we kept training a randomly initialized model (APPO) for a larger number of steps. Indeed, results presented in Appendix F, paragraph “Training from scratch with more steps”, show that a model trained from scratch manages to outperform pure fine-tuning. However, after some time the training stagnates and in the end falls significantly short of fine-tuning with additional knowledge retention (e.g. APPO-BC).

---

> > ### Comment · Reviewer_dWac · 2023-11-22
> >
> > Thank you for the clarifications and additions to the paper.
> > I am willing to revise my score upwards.

---

### Author Response · Authors · 2023-11-15
**General response**

We would like to thank the reviewers for their extensive feedback. We are happy to hear that the reviewers praise its practical importance (#dWac05, #bshN31, #5ZRC30), the scope of the experimental results  (#dWac05, #BuFP04, #5ZRC30) as well as our visualizations (#BuFP04, #dWac05, #5ZRC30).

At the same time, the reviewers raised many critical points, interesting insights, and suggestions. We are also grateful for these! As a result, we have introduced numerous changes to the manuscript marked in the revised pdf.

# Scope and contribution

We start by clarifying that our contribution is *conceptualizing and measuring that transfer from a pre-trained model can be severely limited by forgetting* (dubbed forgetting of pre-trained capabilities). Such effort has the following features:
- **It is non-obvious.** A common practice when fine-tuning from a pre-trained model is to train without additional restrictions, possibly motivated by a view that “plasticity” will promote transfer [1, 2, 3]. We show that in RL context such a view is not necessarily well-founded: there exists a non-trivial trade-off between transfer and forgetting, and, crucially, the control of the latter is needed to retain knowledge useful for transfer.
- **It is important.** The rise of strong foundation models coupled with RL's infamous hunger for computation and data, drives the contemporary, and most likely future, RL research towards training protocols that fine-tune pre-trained models. Consequently, studies, such as ours, that highlight the potential pitfalls of such a procedure and provide ways of addressing them will have an increasingly important impact on the field.
- **It is a fundamental research direction within the RL field, separate from continual learning (CL).** We concentrate on fine-tuning of pre-trained models in RL and *we do not aim to make a contribution to CL*, see the next paragraph for more details.

To the best of our knowledge, this comprehensive approach examined in our paper has not been properly explored in the existing literature.

**Relation to continual learning (CL).** To add to the above remark and make the distinction clear, let us observe that typically in CL one trains the agent on n environments/tasks in sequence: first on task 1, then on task 2, and so on. In our setup, we train the agent on one environment, where each trajectory can comprise all n tasks. Using Montezuma’s Revenge as an example, CL would train the agent on the first room (e.g., for 1M steps), followed by training on the next room (e.g., for 1M steps), and so on. In our setup, we train the agent to solve the entire game, i.e., to solve all the rooms needed to finish the game in one episode. As such, we show that forgetting is harmful even when the downstream task is stationary.

We acknowledge that our presentation has been unclear in the original version of the paper. To address this, we rewrite the Introduction and Related Work sections to explicitly incorporate the points mentioned above. The changes in the revised version are highlighted in blue.

# New experiments
We thank the reviewers for their insightful suggestions that led us to conduct the following additional experiments:
- We conduct studies of different distillation methods in NetHack, see Appendix F. We find that even subtle differences can lead to significant performance loss.
- We examine how representations and policy impact the transfer by performing fine-tuning with a resetted policy head, see Appendix D.
- We check different pre-training schemes on Montezuma’s Revenge to verify the robustness of our conclusions and we conduct an analysis of room visitation, see Appendix E.
- We perform experiments with different sequences in the robotic manipulation benchmark, see Appendix D.
- We check the impact of the memory size on the results of BC, see Appendix D.
- We check the distribution of Sokoban scores to better understand the behavior of the pre-trained model, see Appendix F.

[1] Improving language understanding by generative pre-training \
[2] BERT: Pre-training of Deep Bidirectional Transformers for Language Understanding \
[3] An Image is Worth 16x16 Words: Transformers for Image Recognition at Scale

---

### Author Response · Authors · 2023-11-21
**Summary of the second revision**

We would like to again thank the Reviewers for their comments and suggestions that led to numerous improvements to the paper. Since the last revision, we ran additional experiments and we updated the manuscript to include them.

For clarity, changes introduced in today’s revision (21 Nov) are highlighted in green, while changes introduced in the previous revision (15 Nov) are still highlighted in blue.

We ran the following experiments:
- We verified the performance of the episodic memory approach on Continual World.
- We evaluated EWC on Montezuma’s Revenge
- We evaluated EWC on NetHack
- We ran experiments on NetHack with a higher number of training steps (4B instead of 2B).

As the discussion period draws to an end, we would like to gently ask the Reviewers to let us know if they have any further questions after the rebuttal and to consider raising the scores in light of the introduced improvements. We greatly value your feedback.

---

### Meta-Review · Area_Chair_JHjp · 2023-12-06

**Metareview:**

The submitted paper investigates two potential pitfalls for fine-tuning RL models, which the authors refer to as "state coverage gap" and "imperfect cloning gap". In both cases, forgetting of the pre-trained capabilities can happen, resulting in inefficient learning. Several solutions for counteracting forgetting coming from continual learning and behavioral cloning are investigated as remedies to forgetting and shown to alleviate the issue.

Strengths: The addressed problem (transfer of pre-trained capabilities in RL) is important and the submitted paper does a good job of characterizing two cases in which naive fine-tuning strategies can fail, even in very simple settings. Furthermore, the paper illustrates that applying techniques from continual learning or BC regularization can effectively enable fine-tuning. The experimentation is in general very thorough.

Weaknesses: The obtained results are not really unexpected (I saw the authors' note saying that they are but I don't agree and consider their argument to be only true if one directly considered supervised/unsupervised transfer learning and ignores findings from continual learning). The solutions used to mitigate the forgetting of capabilities are all existing techniques.

The paper received mixed reviews (2x reject, 1x borderline accept, 1x accept). Unfortunately, not all reviewers reacted to the authors' responses but I tried to consider whether the rebuttal would have resolved these reviewers' concerns. While the rebuttal certainly did so to some extent (adding new baselines and other new results), I think it didn't overcome concerns regarding the significance of the paper in the light of existing work. Thus, while the paper is a good read, it lacks significance and novelty and I am thus recommending rejection of the paper.

**Justification For Why Not Higher Score:**

While the experimental evaluation in the paper is thorough and the considered problem relevant, the obtained insights are not very deep, mainly to be expected and can be well explained and resolved using existing methodology.

**Justification For Why Not Lower Score:**

N/A

---

### Decision · Program_Chairs · 2024-01-16

Reject